

# The ethics of using satellite data to monitor and publish research on geohazards in regions of political complexity

Jessica A. Payne[1] and Kate Donovan[2]

[1]COMET, University of Leeds, Woodhouse, Leeds, LS2 9JT, UK
[1]Edinburgh Climate Change Institute, University of Edinburgh, Edinburgh, EH1 1LZ

**Correspondence:** Jessica A. Payne (eejap@leeds.ac.uk)

**Abstract.** Over the last ten years, open-access earth observation satellite data volumes have grown exponentially. Through use of this data, geohazard remote sensing scientists' abilities to impact policy and reduce societal vulnerability to geohazards has become increasingly tangible. However, much of this open-access data is acquired, stored, and disseminated by the Global North, whilst most disasters take place in the Global South. Furthermore, many societies are imaged by these earth observation

satellites perhaps without knowing these satellites exist. There exists little ethical guidance for remote sensing researchers in navigating ethical risks that may arise during geohazard research which uses satellite data. We explore these ethical consider-ations with UK based geohazard remote sensing scientists through thirteen interviews and three focus group discussions. We discuss with scientists their thoughts on ethical review systems for geohazard remote sensing; the importance of visiting study regions; how data should be handled which images people and societies; the desired impact these scientists want to achieve; and

the ethical risks of open data. We find among remote sensing researchers in industry, government organisations, and academia a deep desire to contribute to the reduction of societal vulnerabilities to geohazards. In accordance with geoethical practice, all participants are driven to produce world-leading research outputs to contribute to disaster risk reduction. However, we find among the academic geohazard remote sensing community a misunderstanding around the purpose and need to consider re-search ethics, focusing mainly on personal privacy rather than ethical risks associated with their own research. This community

additionally struggles to balance their desire of meaningfully reducing societal vulnerability to geohazards with the demands of academic expectations and national excellence frameworks. Resultingly, despite best intentions, time, funding, and academic pressures sometimes drive researchers to seek and publish data and results without consultation with in-country communities, particularly post disaster. We recommend geohazard remote sensing researchers receive training to promote the need and ability to reduce ethical risks in their research through five main topics, including practices for decolonisation of geohazard research.

We propose a light-touch ethical review system at project design stage would allow ethical risk identification and mitigation. Our findings and recommendations are in accordance with geoethical practice and build on the Sendai Framework for Disaster Risk Reduction.



# 1  Introduction

Satellite remote sensing is driving the collection of environmental data across the world without the researcher needing to
engage with the local context or indeed seek any ethical approval. This apparently extractive science is commonplace and
vital for a wide range of sciences and decision making including the understanding of land movement, monitoring biodiversity
change, and monitoring hazards that can threaten lives and livelihoods. However, by improving our scientific understanding
we are enabling access to sensitive data and information especially in complex settings. With the constant push for open access
information, this paper discusses the ethical dilemmas encountered when using satellite data to study land subsidence in a
region that is politically complex.

In previous work, we identified 108 rapidly subsiding areas in a country of study. Some of the fastest subsidence rates in the
world (> 150 mm/yr) impact the country's most densely populated regions ( 10,000 /km$^2$). In line with current good practice
UNESCO (2021), we have constructed an open-access, online subsidence portal to share our dataset. We define open access
according to the UNESCO definition of open science in their Recommendation on Open Science (UNESCO, 2021), in which
they suggest "open science encourages scientists to develop tools and methods for managing data so that as much data as
possible can be shared as appropriate".

Due to the political context of the study area, it is not possible to name this country in this study. Moreover, it is not possible
to visit and it is difficult to involve local voices in our portal and data dissemination design process. In a broader context,
acquisition and access to environmental satellite data has accelerated in the last decade, particularly with the launch of the
first European Space Agency's Copernicus Space Sentinel satellites in 2014 (Jutz and Milagro-Pérez, 2020). These satellites
provide online, open-access, medium resolution imagery which is acquired repeatedly on a near-weekly basis for some satellites
in some locations (Sentinel-1 and Sentinel-2 families, (Torres et al., 2012; Main-Knorn et al., 2017)). Moreover, with ever
enhancing compute power and storage capabilities, satellite data processing rates and quality have accelerated alongside data
availability. Therefore, researching inaccessible geohazards on a continental scale is now easier than ever before. Here, we
define geohazards following Tomás and Li (2017) as: "events related to the geological state and processes that pose potential
risks to people, properties and/or the environment, which can be classified within two main categories: natural hazards (such
as earthquakes, landslides, volcanic eruptions, tsunamis, and floods) and human-induced hazards (such as land subsidence due
to groundwater-extraction, water contamination, and atmosphere pollution)".

However, in a conventional disaster risk research setting, geohazard communication focuses on enabling authorities to con-
vey hazard and risk information to those at risk. More recently, emphasis has been placed on disaster management forging
partnerships with local communities to better enable risk reduction and resilience building (Stewart et al., 2023). At the same
time, many funders including the World Bank (Bank, 2012) and UK Research and Innovation (UKRI Fathallah, 2022) have
stipulated any research that they fund should be published open access immediately. As UKRI is a funder of several geophysi-
cal and earth observation projects in the UK (UKRI, 2022; BGS, 2019), this policy instructs all findings relating to geohazards
be published open-access. The only reasons given to protect access to data in UKRI's policy focus on commercial confidential-
ity, use of third-party materials, and potentially identifying human participants. This current open access approach leaves little





scope or guidance for other areas of sensitivity such as research that investigates illegal activity (e.g. (Fakhri and Gkanatsios, 2021) focusing on monitoring illegal fishing) or research focusing on conflict regions (e.g. (Kurekin et al., 2019) aiming to detect post-conflict damage in Mosul, Iraq). The question arises: How do we balance the desire and drive for open access with 60 safeguarding imaged societies, particularly when communicating geoscience to the wider community?

Ethical considerations of satellite remote sensing are explored in great depth by Wasowski (1991). Their reflections recount a history of remote sensing for military and non-military uses, before exploring the ethical considerations implied by the 1986 UN "Principles Relating to Remote Sensing of the Earth from Outer Space" (UNOOSA, 1986). They comment that the first space state-based remote sensing efforts were non-military, with the US formally announcing its intention in 1955 to begin an Earth 65 satellite programme. Despite no international agreements relating to satellite remote sensing preceding this announcement, not one nation protested the planned programme, with the intention being met with widespread approval. In contrast, when President Eisenhower proposed the US' *Open Skies* policy in 1955- a military reconnaissance programme- the Soviet Union were extremely vocal in objecting to space-borne reconnaissance. Such objections were based on the assumption that "any gathering by any means from any altitude of any information which they desired kept secret was automatically espionage 70 and therefore illegal" (Wasowski, 1991). Simultaneously to these objections, the Soviet Union launched the first orbiting non-military satellite of any nation- Sputnik- in 1957, with other states launching their own non-military missions soon after. Again, no notable objections to these missions were raised. Thus, due to a long period of unchallenged practice it became *customary international law* that any nation had the right to overfly any other State with a satellite without prior permission. However, some claim that if they had been better informed, they may have strenuously objected to this space activity (Hingorani, 1988).

At the same time, discussions in the United Nations General Assembly (UN GA) as early as 1958 indicated concern to almost all nations of outer space exploration technology. In response, the UN GA established the *ad hoc* Committee on the Peaceful Uses of Outer Space (UN COPUOS) in late 1958, which became a permanent standing committee in 1962. Two of the UN COPUOS' first documents to be adopted in 1963 and 1966 fail to mention military satellite reconnaissance or non-military observational satellites (Wasowski, 1991), leaving open both the ethics and legal frameworks and implications of using 80 non-military satellite remote sensing systems. This omission led to significant debate on which *Outer Space Treaty* articles to apply to remote sensing, and the establishment of the "Working Group on Remote Sensing of the Earth by Satellites". This group moved to formulate an international agreement on non-military satellite remote sensing. In 1975, UN COPUOS met in session to discuss three draft proposals for such an agreement, with each differing most significantly in the rights of states to acquire and dispose of data obtained from remote sensing satellites. For example, one of the draft proposals put forward 85 jointly by Argentina and Brazil stipulated that: "State parties shall refrain from undertaking activities of remote sensing of national resources belonging to another State party". This suggestion is in sharp contrast to that stipulated by the United States who suggested "every State has the right to sense every other state from space without prior consent, and that all sensed data should be made openly available in the international public domain" (Wasowski, 1991). Despite these contrasting perspectives on satellite data acquisition rights, there was growing agreement, according to Wasowski (1991), that in fact the dissemination 90 of information rather that the acquisition of data posed the greatest threat. Such concern is evident in the eventual consensus reached in the UN Principles. (Wasowski, 1991) reflects on the ethical considerations implied by these principles in significant





detail. Principles X and XI are particularly relevant to the context of this article. Principle X states that: "Remote sensing shall promote the protection of the Earth's natural environment", whilst Principle XI states that: "Remote sensing shall promote the protection of mankind from natural disasters". These principles dictate that any State with information gathered by satellite

remote sensing that could avert environmental damage or be useful in States affected or likely to be affected by so called "natural disasters", respectively, should disclose this data to States concerned. For Principle XI this should be as soon as possible.

In other previous work, Fisher et al. (2021) explore ethical dilemmas for satellite remote sensing researchers. Their academic context focuses on archeology in the Middle East and North Africa but their reflections are also applicable to the field of

geohazards, given the commonality in methodology in both fields. They state: "There is little guidance for conducting remote sensing surveys by archeologists", which they explain leaves the burden of ethical choice with the non-national researcher, a choice which is guided very weakly by existing legislation and guidance. They go on to explain how remote sensing circumnavigates the requirement for engagement with a local regulatory framework which, when conducting fieldwork, researchers must comply. During remote sensing studies, the non-national researcher sits outside of these laws and guidance and has greater

freedom of access to the region. This can, Fisher et al. (2021) describe, create a distinct power shift towards the non-national researcher which can accentuate their ethical responsibilities further. Additionally, this power shift can bring risks associated with digital colonialism (Fisher et al., 2021): the Global North has greater resources for remote sensing research and, to some extent, may exercise imperial control on regions of the Global South through software, hardware, and network connectivity, which could give rise to domination in the remote sensing academic field (Kwet, 2019).

Since Wasowski (1991)'s piece on ethical aspects of remote sensing, over thirty years have passed. Now, the number of observational satellites, volume of remotely sensed data, and activity of remote sensing scientists and agencies is inconceivably larger than in 1991. Many of the theoretical ethical considerations that they identified are playing out with increased intensity as data volumes grow. In particular, as the inequality in compute power and available funding grows between and within nations, divisions between developing and developed nations and societies in all contexts of satellite remote sensing is growing.

In this work, we begin to build, update, and assess the current state of the ethics of geohazard remote sensing raised by Wasowski (1991) and Fisher et al. (2021). We conduct one-on-one interviews and focus group discussions with satellite remote sensing researchers to assess the state of knowledge and concern for ethics in satellite remote sensing of geohazards. Particularly, we focus on research conducted by these researchers using satellite data in regions of the world which are politically or socially challenging from a UK perspective.



## 2 Methodology

To explore these ethical considerations, we conducted a qualitative study using two approaches: semi-structured interviews and Focus Group Discussions (FGDs), both with UK remote sensing experts. In both cases, we aimed to explore how remote sensing scientists based in the UK use, analyse, and translate data and research derived from satellite data.

We use a grounded theory approach to develop the research methodology, defined by Cutcliffe (2001) as: "a theory that is induced from the data rather than preceding them". In this approach, we collected and analysed data simultaneously, with analysed data informing our choices in the next data collection stage, a method referred to as constant comparative analysis (Glaser and Strauss, 1967; Charmaz, 2015). We choose this approach as there is little existing literature exploring the ethics of satellite remote sensing from the view of remote sensing scientists. As grounded theory is an emergent approach, we use data to develop an understanding of ethical considerations without applying preconceived theoretical frameworks. Unexpected theories are therefore more likely to emerge that challenge researchers' preconceptions and representation of research participants (Charmaz, 2015). In this vein, we adopt the constructivist revision of grounded theory (Bryant, 2002, 2003; Charmaz, 2000, 2006; Mills et al., 2006), in which researchers focus on engaging in reflexivity- scrutinising the research process and situation which may influence data collection, analysis, and interpretation. This self-reflective approach is particularly important given the involvement of the authors in the same UK remote sensing community as many of the interview and FGD participants, an involvement which may provide the authors with preconceptions around remote sensing ethics.

Indeed, we used grounded theory to inform our data collection strategies. We iteratively interviewed, analysed, and reflected on the developing themes emerging from interviews and later FGDs. These themes informed the framing of FGD themes and questions. Furthermore, this iterative approach informed the sampling of later interview participants.

### 2.1 Participant Sampling

#### 2.1.1 Interviews

We selected participants for one-on-one interviews using a targeted sampling approach in which participants were identified based on the following criteria. Targeted participants must have: 1. been an employed member of staff in the UK; 2. had current or previous experience working with satellite remote sensing data in the context of geohazards. We aimed to recruit participants from across the UK remote sensing spectrum, aiming, where possible, to diversify participant gender, career length, and career stage.

Participants optionally self-identified their age, nationality, gender, the current and previous sector(s) they worked in, the current and previous role(s) they have held, and the number of years they have used satellite data in their job or career. This information allowed for a qualitative assessment of, for example, if career stage or gender impacted participants' opinions.

All of the initial seven interview participants were members of The Centre for Observing and Modelling Earthquakes and Volcanoes (COMET), a UK-scale community of world-leading scientists who study earthquakes, volcanoes, and other geohazards using remotely sensed data. These initial interviews were conducted May-June 2023, with interview results used to build themes and questions for the Focus Group Discussions (FGDs) held at the COMET Annual Meeting 2023 on $26^{th}$ June,





2023. A further four interviews were conducted soon after these FGDs in June-July 2023 with the same question structure as the initial seven interviews. These later interview participants were selected to diversify the mix of interview participants

from industry/academia/research organisations. Finally, after initial coding analysis of these first eleven interview transcripts, a further two interviews were conducted in February 2024 to further test emerging hypotheses relating to female participants and those employed in industry.

We chose to focus on scientists in the UK to purposefully limit the scope of the study. This choice allows time for a more complete exploration of remote sensing scientists or professionals whom, for example, work in an industry and academic setting

that experiences the same legislative framework, similar access to satellite datasets, and most likely experience a similar world view. Additionally, focusing on only UK remote sensing scientists made it more likely that we would be able to recruit a representative mix of participants, given that two authors were already members of the UK remote sensing community.

### 2.1.2 Focus Group Discussions

FGDs were conducted in person at the COMET Annual Meeting 2023 held at the British Geological Survey Keyworth site.

Meeting attendees were given the option to partake either in this study's FGDs or one of two other 'Breakout Group Discussion' sessions which were held in parallel. These parallel sessions were allocated 50 minutes. Meeting attendees were notified of these breakout session options via email one week before the FGDs took place, but attendees had up until the start of the parallel breakout sessions to opt-in or out. Due to this self-selecting approach, there was a strong diversity to the FGD participants. Additionally, as we did not restrict the FGDs to staff only, as we did for the interviews, the diversity of career stage and age was

greater than that for the interviews. This approach we hoped would widen the diversity of voices within each focus group, thus generating a greater diversity of opinion and increase the likelihood of differing opinion between group members, enabling us to analyse how focus group members dealt with differing ethical considerations for satellite data science.

Overall, twenty-two COMET Annual Meeting 2023 attendees self-selected to take part in the FGDs. We split this group randomly into three smaller groups to enable discussion. Focus group one had seven participants, group two had eight, and

group three had seven. All participants except one in group one submitted a Participant Information Form and Consent Form meaning we lack personal identifying information for this participant. Verbal consent, however, was additionally gained from all participants. A detailed breakdown of FGD participant information can be found in Figure A1.

### 2.2 Data Collection

Following grounded theory, we conduct data analysis during and alongside data collection to test developing theories. As the

same person planned and conducted data collection, data transcription, and data analysis throughout this study, the same author viewpoints, subjectivity, and preconceptions are applied at all stages. Thus, exploring and reflecting on researcher positionality and potentially incorporated preconceptions is more straightforward (Charmaz, 2015). Moreover, this researcher is more likely to pick-up on nuances revealed by participants during interviews and FGDs as they were present throughout.



### 2.2.1 Interviews

We conducted thirteen semi-structured one-on-one interviews ranging in length from 37 to 38 minutes with a mean length of 49 minutes. The broad interview structure was based on five main sections: 1. Participant background questions; 2. Regions that the participant has conducted research in or on that are challenging politically or socially; 3. Ethical implications of using satellite data for research; 4. Land subsidence specific questions; 5. Translation and communication of satellite remote sensing outputs. Each section was broken down further into sub-sections depending on the participants' research experiences. For example, we initially asked participants if they have ever researched an region that is difficult to visit due to social and political reasons. If no, we moved to a new subsection which explores which regions in general they have researched. Each sub-section contained a list of questions that were a mix of closed and open, with question choice depending sometimes on participants' previous answers.

This questioning structure allowed consistency between interview approaches and provided structure and direction for participants. However, following the grounded theory approach, if topics arose which participants appeared to define as crucial and important, these would take precedent over the planned interview structure (Charmaz, 2015). Equally, if a participant was not responding well or openly to the planned line of questioning, the interviewee moved on to topics which the interviewee sensed would ease the participant into the interview environment, thus changing the planned interview structure.

Participants were given the option to conduct the interview online or in-person. Due to four participants working elsewhere in the UK to the University of Leeds, these interviews were conducted online. Eight of the other interviews were conducted in person at the University of Leeds whilst one interview was conducted the British Geological Survey Keyworth site. In-person interviews were recorded using Audacity Version 3.3.2 (Team, 2023). Online interviews were recorded using either Zoom (Zoom, 2023) recording facilities or Microsoft Teams' recording facilities depending on which software the interviews were conducted.

### 2.2.2 Focus Group Discussions

We conducted three FGDs in parallel with discussions led and guided by moderators. FGDs were designed to confirm or cross-check findings developed during the initial interviews.

FGD topics were chosen based on questions in initial interviews which led to the most varying opinions from interview participants. Topic 1 focused on remote sensing and ethical review. In interviews we asked: "Research involving human participants requires ethical review. Research that observes humans using satellite data of any spatial resolution does not. What do you think of this statement?". 25% of initial interviewees thought there should not be an ethical review process for satellite remote sensing in a scientific context; 45% had a mixed opinion; whilst 30% were in favour of an ethical review process. Given that opinions were mixed not only between but also for individual participants, we were interested in how group members would interact on this question and whether participants views would develop through group discussion.

Topic 2 explored study region access. We asked in interviews, for example: "If a scientist cannot visit the study region, should they still research it [using satellite remote sensing data]?". Again, this was a contentious topic with interview responses



tending to focus on individual experiences, their projects, and their success using only remote sensing data. We therefore wanted to explore whether participants disagreed or agreed with each other on these opinions, and whether participant age and career experience impacted this opinion.

Finally, Topic 3 explored the definition and the ambition of making an 'impact' by satellite remote sensing scientists who research geohazards. In initial interviews we asked: "How would you define making "impact" when researching geohazards using remote sensing?". This question aimed to explore the motivations of these scientists, whether they were embedded in advancing scientific theory, protecting civilians, or otherwise. Initial interviews highlighted that "impact" to remote sensing scientists who researched geohazards ultimately meant contributing to protecting people and saving lives. We wanted to see if 225   this ambitious aim remained during a focus group discussion and if participants challenged each other on their desired impacts.

## 2.3   Data Analysis

Following on from our grounded theory research method, analysis was conducted in a comparative, iterative, interactive way (Charmaz, 2015). Firstly, in order to extract both explicit and implicit meaning from our data, all interview and FGD recordings were transcribed verbatim. Resulting transcripts provided a rich dataset from which to illicit thoughts, feelings and intentions 230   from participant responses that may not be surficial. It is worth noting that some argue that transcribing wastes time and risks becoming lost in data (Glaser, 1998).

     We use NVivo 14 (Lumivero, 2025) to label interview transcripts with codes. Given the same researcher conducted all interview questioning and transcription, as well as coding, the author began with a small list of initial codes developed during data collection and transcription.

Initial coding involved using broader themes based on interview questions including "International guidelines", "Collaboration", "Illegal activity", "Ethical review", and "Visiting study regions". More specific codes were developed once interviewee transcripts were divided among these broader themes. This latter coding approach took verbatum lines now defined as being part of these broader themes and elicited from them further meaning focusing on sentiment and meaning. These latter codes included "fearful of ethical review", "desires ethical direction", and "interprets ethical review as yes or no process".





## 3  Results

As the interviews and FGDs were conducted in a small time window of ten months, the results represent a snapshot of opinions and considerations of the ethics of geohazard remote sensing. Due to the focus of this paper on the ethics of remote sensing, we discuss in our results responses to topics 'Ethical implications of using satellite data for research' and 'Translation and communication of satellite remote sensing outputs'.

### 3.1  Ethical review for geohazard remote sensing

We asked interview and FGD participants: "Research involving human participants requires ethical review, such as this interview, but if you research observes using satellite data at any spatial resolution, you don't need any sort of review into the ethics of it. What do you think of these statements?" We did not explain the process of an ethical review and we left it up to the participant to reflect on what ethical review means to them. Around a quarter of interviewees responded favourably about the implementation of an ethical review for satellite remote sensing of geohazards, one participant was against, and two-thirds had a mixed view. In focus groups, views were mixed but in general, all three groups agreed that ethical review should be considered under some circumstances.

There was strong focus by interview participants on the ethically positive nature of their work on geohazards to ultimately contribute to reducing risk to geohazard impacts. The one participant vehemently against ethical review was fearful of how ethical review could limit environmental research that would otherwise increase awareness and preparedness of local people to under-researched geohazards. They said: "Once we say the people benefit from research, that is the main benefit...That is the main point. You should not let any other criteria affect this main objective". Other participants with a mixed view ethically supported this viewpoint that geohazard research, using whatever methodology, is ethically favourable. For example, one Senior Academic who researches earthquakes suggested that their research ultimately "makes the planet more resilient to earthquakes...There is an overarching ethical reason why we are doing this". Similarly, several participants suggested that, if you are simply researching a geological or geophysical phenomenon then ethical review may not be required. However, many felt that once you identify a potential hazard to society then you need to start to consider ethical questions. For example, one participant suggested that if you are researching slip vectors on a fault then this has no hazard implications so does not require review. However, this same participant knew a colleague who identified strain accumulating on a fault two years before it ruptured, the shaking of which lead to thousands of deaths. This colleague "felt incredibly bad" about not communicating better their findings and ended up working for the UN on disaster response partly to "stop the demons". This participant asks: "We write lots of papers like that [on strain accumulation], this doesn't mean you shouldn't do the research. What could he have done? Should there be an obligation to work with local partners so that information is shared for future hazard mapping? That's where [ethical review] is worth thinking about".

Those with a mixed view towards ethical review mentioned study and data context, administrative burden, and the remote nature of satellite data as items that influence their thoughts on ethical reviews in geohazard remote sensing research.





In relation to study specific variables, a third of participants across academia, industry, and government organisations mentioned data spatial resolution as a consideration as to whether a review is needed. For example, one participant who is a Senior Academic said: "I think I don't know. I can see where the motivation for that [review] comes from and I can see that if satellite

data is of sufficient resolution to detect, for example, illegal activity and someone might get in trouble when you publish it, that's something on your conscience...On the other hand, if you can't do the science and publish the science in an open access way because of that risk if the risk is very low, then I think that would be a shame. The resolution for most of the imagery I use at the moment, I don't see that these things could really be picked up". Here this participant is referring to the use of SAR data such as Sentinel-1 (maximum 5-20 m spatial resolution), TerraSAR-X (1-3 m), and COSMO-SkyMed (1-100 m).

Despite this participant being a Senior Academic in the field of satellite remote sensing of geohazards, they are conflicted or unsure if an ethical review for their research should be applied or implemented. Another participant employed in an industrial setting again mentioned resolution as a motivator for ethical review, saying: "with very high resolution [data] becoming more and more accessible...it becomes higher and higher risk...More people are sort of saying, OK, we need to be careful with it". The participant describes the decision of completing an ethical review for a project to be up to the Project Manager themselves-

they decide whether a project should be assessed by in house lawyers, for example, to assess any legal or ethical issues. This participant describes that there has been a shift in industry in the last (before 2023) two to five years in the sector towards becoming more formal in the approach to filling out ethics forms during satellite remote sensing projects.

Focus Group 3 also tended to focus on the ethics of privacy and individual identification. These participants discussed that humans are not visible in satellite data that geohazard scientists tend to use and thus, they implied, there is no need

for ethical review. Another participant supported this notion suggesting, even if a researcher could see a cluster of people in a satellite image, that a researcher cannot recognise them or identify them, negating the need for ethical review. A third participant countered these suggestions saying: "But you can see paths actually. After some eruptions you can see where people have marched in certain regions, so can track people's motion", highlighting that there is some identifying information to be derived from satellite imagery. Group 2 also discussed resolution as a consideration for ethical review. Ethical review

should particularly be considered if people were captured, one participant said: "doing certain behaviours". These behaviours were not defined, however. Another participant compared the existence of regulation surrounding drone footage taken over, for example, a residential property to the lack of regulation for similar imagery— albeit at lower spatial resolution— acquired by satellites. This comparison was used to support the need for ethical review of the use of high resolution satellite imagery in academic research.

Considering the socio-political context of a study's target seems important to some participants in whether an ethical review should be conducted. One participant employed in a GRO said: "it depends what you're looking at...if you're looking at how people are moving in a crisis situation like refugees or something like that...you need to consider the ethics of that in terms of how open that data is and who might be interested in that data, especially if those refugees are from a war". When asked to expand, this participant described that there may be groups persecuting these refugees who are interested in where they are

displaced. Therefore protecting these refugees should come above openly publishing any findings relating to their location. Another participant employed in academia also discusses this issue of conflict but in the context of time sensitivity of conflict





related information. They discuss that, working on a war-impacted region using satellite data would bring ethical concerns if satellite data and results were posted online, particularly if this was done "really quickly and dynamically". They suggest, for example, that using such data with a time lag after acquisition such as five years in the future would be less of an ethical issue.

In further regard to the subject of geohazard remote sensing research being important in considering whether to complete an ethical review, one participant explained the care that some researchers already take in a disaster-context. They said: "Around the Turkey earthquake (Turkey-Syria earthquakes, 2023 e.g. Naddaf, 2023) a lot of people did have care, a lot of researchers were like: "we are not going over there, we are staying out of the way" this was when they were going to do fieldwork. Even remote sensing people were trying to keep up links to Turkey and Turkish researchers and trying to be very careful about

standing on their toes. But in a way that wasn't a requirement, there was no ethical regulation there they just thought that was the best. So if there is a space for it [ethical review] is in those regions [of research]".

    Some participants suggested adding ethical review to a geohazard satellite remote sensing project adds paperwork and consumes time. In general though, these participants did also feel that the need for a review may now outweigh this administrative burden. One Senior Academic said: "I don't like filling out forms, ethical reviews take more time, but as the resolution in-

creases, it is starting to concern me more and more about what we do and how we release it". An employee at a GRO says that the documentation is "quite tedious" and that they would prefer to keep and simplify the process rather than get rid of it.

    Some participants touched on the non-intrusive nature of satellite remote sensing as reasons both for and against ethical review. One participant employed in industry said: "I think that, yeah. I think it's actually because people don't know that the data is necessarily being collected about them". They compared this non-intrusive nature of satellite remote sensing to air

quality data sampling, which sometimes requires hand-held measurements on the ground. The participant suggested that being able to physically see a sensor recording data near a person was more ethically compliant than the non-apparent nature of satellite data acquisition. One participant informant conversely suggested that once you are outside your house you have no privacy anyway, so those being observed by satellite data shouldn't expect a review.

    A participant employed in a GRO provided insights into their ethical review system. In their case, an ethical form is not

required for every project. Additionally, the review focuses more on data storage and adhering to, for example, GDPR regulations rather than the ethics of geohazard satellite data related research. Within this process, the participant says: "you [the researcher] need to understand if what you are doing is good or bad". When asked if they understand well enough if their research is "good or bad" the researcher says: "probably not sometime...especially when we consider these places are not Western society...Rather than trying to replicate what we do here, sometimes it's important to have those...personal connections so

they can advise and guide you". For context, this participant conducts research into geohazards mainly in Southeast Asia. This participant's reflection is important and highlights that the ethical values and views of "good and bad" in a geohazard remote sensing research context are shaped by the values of the societies and groups that we are engaged in. Imposing such ethical views and practices on those in a different societal context may not be appropriate

    Focus Group 1 participants were in favour of an ethical review process for geohazard remote sensing to better consider

the community impacts of research. One participant said: "You never know if you are affecting other people, or groups, or communities. So I think always you need to review ethically". Following on from this suggestion, two other participants




agreed that precedence should be applied from other disciplines. One participant said: "It think it would be interesting to see what are the conditions in social science research for ethical review, where it is necessary, where it isn't. Not all social science studies require ethical review...We should think carefully about how those apply to satellite imagery. If it's anonymised, for
example, maybe it's less applicable, for example". This final suggestion further implies a key ethical issue for study participants relating to satellite data is it's potential to identify individuals.

### 3.2   The importance of visiting a study region

We asked interview participants: "If a scientist cannot visit the study region, should they still be able to research it?". We asked this question to explore participants' views on the role of satellite remote sensing in the global research community. When
asked this question, many participants began to consider the role of collaborators in research as collaboration is either essential or simply insightful when conducting research in another country. Overall, two-thirds of interview participants thought that researchers should be able to research a region that they have not visited, with the majority of the rest having a mixed viewpoint. All three focus groups' discussions around this question were polarised, with stronger opinions in favour of visiting a study region than in interview responses.

Many participants agreed with the notion that, as satellite remote sensing data on a near-global scale is free and openly available, researchers are then able to— and should continue to— use this data to conduct research anywhere that data exists even if they cannot, or have not, visited the study region. One participant from a non-academic setting said: "The beauty of remote sensing is that you can study pretty much anywhere in the globe. If I didn't say yes I couldn't do [my project]". In agreement with this sentiment, one informant who is knowledgeable about a non-UK study region highlights the need for
remote sensing studies to fill geohazard knowledge gaps in data-sparse regions which may lack capacity in country to develop these results themselves: "[Yes] we do not have much information about [the study region]. Anyone from any part of the world, if they are able to produce something, deformation layers, whatever layers, if they can be useful for local government or people of [the country] this would be very helpful." Similarly, focus Group 3 opened immediately with one informant saying: "I am struggling to see the ethical pitfall of using remote sensing data, it seems like if you are doing things from the right place,
remote sensing can be [useful]. If the answer is no then you are actively preventing research where it might need to be done". This informant goes on to give an example where remote sensing is valuable when in-person access is difficult, saying: "There is no way to get data from Iraq so you have to do it all remotely". Another participant implied caution must be taken when visiting a study region, saying: "To go there just to go there is not right, but to go there to collaborate with people is good".

Others who have a longer satellite remote sensing career noted a change in attitudes to visiting study regions and collab-
oration over their career. The participant said: "I think maybe 20 years ago people would just go and look at something and publish it...now it's much more, you know, kind of trying to collaborate with people and trying to have partners from relevant organisations in country." Asked why they thought this change might have taken place, the participant said: "a lot of the research in the past has been done by developed nations, and if you're doing that in a developing nation, then particularly if you were British and you were researching something in India, for example, it might look like you're kind of imposing a kind of
colonial type thing or something like that".





Other participants seemed more aware of the ethical choices that researchers face when considering to research another region completely remotely. For example, one participant said: "I think they can [research a study region without visiting it]...I don't see why they shouldn't...and this is where you come to the problem with remote sensing data...it comes down to whether you should try and establish contacts with people in the country or in the area". This quote highlights the internal

ethical struggle remote sensing researchers can face when deciding their study region. The ethical choice of establishing in country contacts lies with the researcher; they are free to conduct projects completely remotely or spend time establishing in-country partners. Similarly, one interview participant explained that, despite best intentions and attempts to engage with local researchers, sometimes this engagement fails. This GRO participant said: "We are just about to publish a paper on...[a city]...for example with no [city] authors, not for lack of trying. They just don't have time to contribute. And that's it. We have

been trying for four years to get them involved and keep them up to date and all that stuff and ask them questions but ultimately people are just really busy and that feels like extractive science with just 4 [UK-based] co-authors, but the context is that we have tried really hard to keep them engaged and asked them the right questions. We have been there; we have done training programmes out there. They have been here. We have tried to do things as well as we could but it hasn't worked".

Continuing on this theme of collaboration we asked Group 1: "Do you think there are any situations in which you shouldn't

complete the study if you can't access the country?". This group had strong contrasts in opinion on the ethics of remote sensing studies working external to a study region. Immediately, one participant employed in a GRO said: "If you are not in local partnership". When asked why they thought this was important, they said with strong conviction: "Should you be working anywhere without a working partner? I'm not sure what gives us the right to work in somebody else's country". From here this group discussed whether this assumption applies in certain situations with one academic saying: "With remote sensing data,

if there is some kind of...eruption that hasn't happened [before], certainly not in the satellite era, there is a rush for people all over the world to bring their opinions to bear on a new scientific phenomenon. A lot of that work did not wait for people to reach out to people on the ground and say "do you mind if we study the plume $SO_2$" or whatever else". Another participant who's work focuses on earthquakes compared this volcanic example with a similar situation following the Tōhoku earthquake of 2011 (e.g. Kagan and Jackson, 2013). They said: "It was the same situation there where lots of people jump in and say I can

do this I'll look at that, it's a kind of rare event. It's the same idea. Lots of people looking at this data didn't have agreements with Japan, but Japan made lots and lots of this data available, people were able to look at GPS and other data in a way we didn't really need it [the collaboration in-country]".

Focus Group 2 additionally had contrasting opinions around this question, again with many participants strongly in favour of study region visitation to improve scientific collaboration but also researcher awareness of the wider community. "If you want

to understand the physical processes going on, you want to visit the site". This same participant highlighted the importance of ground truth. Providing ground truth data for satellite data means ensuring the measurements estimated using satellite instrument data is accurate through comparison to data collected on the ground. This ground truth process is usually conducted within the study region. Another participant countered these opinions by suggesting that some locations are fundamentally unreachable, with remote sensing being a useful tool where topography is high, or the location is too dangerous to visit. A

majority of this group's participants, however, felt a strong element of visiting a study region was the natural involvement





you have with local collaborators, field researchers, and the wider community when you are physically on the ground. This majority suggested this interaction would benefit collaborators in both directions. For local people, the chances of collaboration and data sharing by the external researchers are increased if they visit the study site. For the external researcher, they benefit from having the chance to learn first-hand how communities are affected by a geohazard. This sub-group of participants were

non-UK national researchers working in the UK. To counter these positive aspects of visiting a study region, a UK-national researcher responded with some frustration, saying: "What happens if you reach out to an observatory or a local science team and they don't want to know? Some real, high quality, useful data exists and what do you do then? In theory, scientific knowledge should be everyone's. But if the principle is that you want to work with local people, this is an excellent idea 90% of the time but sometimes people might not want to do it. Maybe they are working with someone else, maybe they have lots

of reasons. This is a tricky one". As with some of the interview participants, this UK national academic said that the strength of remote sensing is that you can monitor places which don't have capacity to measure signals on the ground saying: "Remote sensing might be the only thing they [in-country researchers] have got".

Some participants suggested the visitation of a study region is required less if you are researching a phenomenon that acts over a long distance in space. For example, we interviewed several researchers who study regional tectonic systems or

estimated strain accumulation on faults over ∼100 km wavelengths. These researchers pointed out that research quality would not improve drastically if they did visit their study region as what they are measuring cannot be measured easily in the field. However, one such researcher suggested that visiting a study region would mean you must interact with local colleagues to, for example, conduct fieldwork or research. This interaction would increase your ability to access local data and knowledge such as local Global Navigation Satellite System (GNSS) data, and thus, to an extent, improve the quality of your project.

Several participants mentioned the importance of satellite remote sensing in mitigating risks of fieldwork and student re- search projects. One participant said: "Using satellite data mitigates the risks [e.g. failed equipment, COVID] associated with conducting research in the field." This participant additionally had limited research budget when conducting their PhD research in an extremely remote, mountainous region. They said: "Logistic restraints makes conducting research more expensive in [this region]. Especially on a, for example, PhD budget. You often don't have the budget to involve local collaborators as much as

you would like". This participant explained that being able to use open-access data provided by the European Space Agency meant that part of their research could be conducted with no cost and meant that they did not have to make the ethical choice of conducting fieldwork but not being able to properly compensate local collaborators. Additionally, this participant explained that using this satellite data mitigates against unanticipated risks such as a global pandemic which may halt any planned fieldwork- open-access data provides a reliable back-up for time-limited projects. Overall around half of this participant's PhD relied on

this open-access data.

## 3.3 Data which images people or societies

We asked interviewees: "In the imagery or data that you use, can you observe people or society in any way?".

Many of our interview participants said that the data that they worked with was insufficient resolution to image individual people or assets in images. Within COMET, for example, many researchers work with medium resolution (100 m) Sentinel-1





Interferometric Synthetic Aperture Radar (InSAR) data which observes the movement of the ground surface at a maximum rate of ~2 m per year. In these cases, participants said that they cannot see people directly. You can see the locations of buildings and settlements indirectly, however, through use of a coherence metric estimated using SAR data. This coherence is higher where the ground surface is more stable, such as the location of a stable building or structure. Conversely, if a building collapses during, for example, strong ground shaking caused by an earthquake, a change in coherence over time can be used

to infer earthquake damage. Another participant described being able to see a settlement destroyed by a pyroclastic flow using optical satellite data. This participant, who works in volcanology, said they had not thought about being able to observe people and societies before and that examples such as the interaction of volcanic flows with settlements was the most stark example from their work.

Some participants said that the impacts of people can be identified with satellite data, rather than the people themselves. For

example, the movement of the ground downwards over time, land subsidence, can be mapped in space and time using satellite InSAR data. Some of this subsidence, participants explained, is caused by humans abstracting and depleting fluids such as groundwater in the sub-surface.

Some participants use satellite data which is more revealing. A couple of participants worked with very high resolution optical imagery, such as that from Airbus' Pléiades constellation. Imagery from these satellites has spatial resolution as low as

~30 cm. When asked what can be identified with this imagery, one participant said: "Over cities, buildings...Not at the level of viewing inside of peoples windows. You can't see registration plates on cars for instance".

Some participants compare the data that they are using to services and imagery provided to the public for free via Google Earth or Maps. For example, one participant who works with Pléiades imagery said: "You do not see more [in our optical data] than you could on openly available datasets like Google Earth". Conversely, another participant who also works with this

optical data explains that Google may actually pixelate or exclude data from either Google Earth or Google Earth Engine if a wide range of people may be at risk. For example, if imagery on Google Earth covers, they said: "strategic infrastructure or war zones", Google may pixelate, for example, an army camp so that "no side stands to benefit" from this imagery being openly accessible.

We also asked participants if they had ever viewed illegal or compromising activity, intentionally or unintentionally, when

conducting research using satellite data. Three participants who were employed in GROs or industry said that projects they were working on specifically aimed to find or research illegal activity in non-UK settings. In an academic setting, one participant said: "Probably yes due to the amount of data we use that something [illegal] has been imaged. This is likely just small scale logging but [a scientist or data processor] would have no idea. Sure there are ways to find out. If you do not know it is there you wouldn't take any measures against this". Two other academic participants thought they had perhaps observed illegal activity

(for example, evidence of illegal groundwater extraction in land surface subsidence) but it would be difficult to find out without local contacts if this activity was within the law. Three academics said to their knowledge they had not viewed such activity.

In response to being asked about illegal activity, one participant explained that some satellite data providers restricted imagery access of places where there is, for example, contested land. They said: "For example the German Space Agency has some countries they just don't let data out because there is contested land...for example it is very difficult to get TerraSAR-X





data of Ethiopia. And I have had data requests turned down on the Ecuador-Colombia border which has been a pain. Because there has been activity or unrest going on around there". The participant speculates that this restriction is related to security sensitivities of these images, perhaps informed by space agency military contacts.

If participants said that yes, they could see people, societies, or the impacts of people in their datasets, we asked whether participants took measures to mask or exclude these parts of their dataset from their published work. No participant said that
they had taken such measures.

We also asked participants if imaged people have a say in the imagery being acquired over them. All participants we asked said no. One participant who works mainly with InSAR deformation said: "these people are not imaged, I would say".

Building on these interview responses, we asked focus groups: "Should remote sensing researchers take measures to keep data that focuses on or images society private?". Group 1 as a whole thought that there are occasions where these measures
would be appropriate. One participant opened the discussion with: "If there are parts of the image you are not using for research this seems a reasonable thing to do...In the same way you mask incoherent stuff [incoherent pixels in InSAR data], you could mask private information". However, several participants countered this suggestion, with one asking how private areas would be identified and another pointing out that, if a researcher is mapping exposure then these human-related features cannot be masked. They said: "That image is fundamentally capturing the infrastructure and things. So there's a halfway house here. So,
yes to the ethics approval, but I think we have to account that the data is freely accessible anyway. And it's there for others to use so if you're deriving a product and doing interpretation, to what extent do you need to apply masking to something someone [could access anyway]"? A third participant counters this suggestion by emphasising the fact that scientists uncover more information from "raw" imagery through analysis and observations. They said: "Is the data itself the part that is ethically problematic or is it your judgment of the group [in the image] in question? You probably shouldn't say "oh the state has
done this", you draw a set of conclusions that lead to these things. The data itself is just pixels". Finally, this group discussed the ethics of the impact that geohazard science derived from satellite imagery can have on house or insurance prices. Two participants provided examples of where hazard maps or damage maps had been used to determine insurance prices. In one instance, the publication of a hazard map in the Azores, one participant said, lead to house prices suddenly dropping. In the second instance, a second participant said that buildings were classed as damaged or not by a geohazard. This damage map
was not accurate and caused difficulties with insurers for homeowners when their house was wrongly classified.

A participant in Group 2 suggested that these measures should be taken in accordance with precedence from existing privacy laws. They said: "GDPR type principles come into this don't they. Can you identify people? You should not publish anything that allows identification of individuals. It is precedence". This suggestion is similar to the precedence logic mentioned by Group 1 in response to their thoughts on ethical review.

A participant in Group 3 suggested that the question of making data private relates to the research intent. They said: "If you are interested in $SO_2$ and you accidentally capture some images in high resolution of people and you're not studying or interested in them, then I don't think for me that would present an ethical problem". Other participants appeared to agree with this sentiment. However, the same participant seemed to reconsider their own point of view when considering Unmanned Aerial Vehicle (UAV) data. They explained that in the UK, if you have a UAV with a camera, even a toy one, you have to





register it with the civil regulation authority. When asked if there is any ethical review process for using a UAV, the participant said: "As an individual no, but as a University as soon as the University realises that there is any consideration then you would have to go through ethical review for sure".

### 3.4   Desired impact

We discussed with interview participants what they feel "making impact" means for them in terms of their work relating
to satellite remote sensing of geohazards. We also asked what dissemination and communication methods they feel are the most effective to achieve impact. For some participants we did not ask this question directly but broadly discussed the topic. To explore the definition of impact further with focus groups, we asked FGD participants directly: "How would you define making "impact" when researching geohazards using remote sensing?" The answers from focus groups, in comparison to interview responses, where more general and less specific to their own day-to-day work. This contrast may be because interview
participants had more time to explain their research and how they personally disseminate research outputs, thus they already began thinking specifically about these questions in the context of their own research.

The definition of making impact and the methods of making impact varied with where participants were employed be it an academic institute, industrial setting, or a GRO. These two items also varied with the geohazard being researched (volcanic or earthquake focus), as well as the region being researched.

Participants employed at GROs, for example, focus on making impact through establishing long-term personal relationships and co-developing scientific results with in-country partners. When asked what is the best kind of impact you want to see when doing your work, one such participant said: "It's that they take over and they start to become independent". For the participant this meant collaborators gaining the ability to process data and publish results themselves to establish research independence. When asked how long it takes to build up relationships to create such impacts, the same participant said: "I
think, at least ten years, I mean we have already some contact, it's still you probably need [ten years]...it's important keeping that relationship [going] so regular meeting, not just limiting to, let's see, one visit every year, [it] is the daily conversation". This participant went on to emphasise the need for personal relationships in having this desired impact, saying: "if someone from [my organisation] who knew that person [collaborator] for 30 years, then he retired or she retired and that's it. You have to start [the collaboration] again".

A Senior Academic who researches volcanoes using satellite data described the contrasting situation in an academic setting. When asked how they define making impact, they said: "Well I think it is all coloured by the UK university definition so we have a clear definition of what impact is here, you write a paper and there's a clear paper trail of what connects the results of that paper to a decision that's made by a decision maker in politics or community civil protection or something like that". However, they then said: "Personally I feel like impact is a bit broader than that. As in, if you establish a long term relationship
with people, and you implant a perspective or provide some different data then that does make an impact on their thinking and that is some kind of tangible results in terms of the way a volcano observatory operates". They explained, for example, that they contributed scientific results that influenced the position of a GNSS site on a volcano. This is a really "critical part of the impact", they said. Asked how they feel about the contrast between these two definitions of impact, they said: "Yeah I think it's





not ideal, I think things that get credit when you have too concrete and too limited a definition of impact are things that have slightly similar horizons and are less ambitious and less risky, and I think it pushes research to be a little bit less adventurous and less exciting if you are focused on trying to demonstrate a specific impact you can actually predict".

Some academics believe that the obvious impact of geohazard research ultimately relates to saving lives through risk reduction. This impact relates to the overall ethical good some researchers feel that goehazard research contributes to society. For example, focus Group 3's discussion mainly focused on the view of one participant who said: "An obvious one is saving lives. You can use data to assist in risk reduction in challenging places. If you have skills in remote sensing you can transfer these to build capacities in countries and allow local scientists to make decisions. Impact in its purest form is reducing risk." Similarly, Group 1 focused on the importance of seeing change in policy or behaviour as having impact. For example, one of Group 1's participants worked on the scientific advisory committee for the UK Government's Foreign, Commonwealth, and Development Office (FCDO). This committee focus on Montserrat and their work goes directly into decision making, impacting the size and extent of the Montserrat evacuation zone. This, the participant said, was an example of making direct impact. Group 1 participants agreed that in general some kind of policy, regulatory, or behavioural change is impact.

For academics who study earthquakes using satellite data, making a desired impact can feel more difficult, according to our participants. One Senior Academic explained that it is more difficult to make impact when communicating research related to earthquake hazard. They said: "it is easier for volcanoes" due to their static nature and existence of observatories. For earthquake hazard, they said: "how do you persuade a mayor of a town who is in the position for three to five years, to do something for a hazard where the risk is actually quite low?...The danger to loss of life is actually quite small, the risk of it hitting a town in a five year window is small but the costs to move the dial are beyond any budget. How do you reconcile that? That's the biggest challenge". However, this same academic appeared to feel a competing tension in making more useful and useable their research, their lack of training in doing so, but additionally a frustration in working with social scientists. They said when explaining a large Disaster Risk Reduction (DRR) project and how it works: "I think there is some interesting work going on there. I'd like to be involved with it but I don't really know, I haven't really got the training. So it's kind of making it up as you go along really". However, when explaining their experience working with other scientists and colleagues who do have that training in DRR, this participant said: "I think that when you start working with social scientists they kind of— depends who they are— whether they want to get things done or whether they are a bunch of social scientists who, if you ask them a question they just ask the assumptions behind your question. If you ask them a question they ask "why have you asked that question?" That is what they are interested about rather than answering the question. So I think it's a bit difficult for people of maybe a scientific mindset when you are trying to solve a problem or do something".

The desired impact and strategy to achieve this desired impact can vary significantly depending on where the researched geohazard is located. One academic researching Iran, for example, explained how the ability to collaborate with international colleagues has changed dramatically over time due to the impact of international sanctions. Before 2017, Iranian universities awarded "bonuses" for collaborations with international scientists. Since 2017–2018 this changed completely with the implementation of international sanctions. International collaborations are now often not offered by European countries, or allowed by Iranian universities. Scientific collaboration can still occur, according to the participant, but with countries such as Russia





that Iran still has a good relationship with. The most impacted collaborations by sanctions are those which the participant says
are most useful, such as those with Canada, USA, and Europe. The participant said that many scientists have left universities
in Iran due to the impacts of these limitations. Due to this situation, the participant said that external countries publishing data
and publications open-access makes science in Iran easier in times of international sanctions. Additionally, when asked the best
method to exchange knowledge as an international scientist with Iranian people or scientists, the participant said that sharing
knowledge in any way researchers can is necessary. People in Iran can access some social media so simplifying results first
and producing easy maps that can be spread through social media should be the first step.

Alternatively, many participants warned against the sharing of satellite data related results on social media, especially if
these results relate to volcanic unrest. Two Senior Academics who have shared data with societies impacted by volcanic events
emphasised the importance of letting volcanic observatories and civic leaders do their job in leading responses to such unrest.
For example, one such academic said: "You also don't want to stand in front of a volcano observatory and say look if it was
me I would call the evacuation off because I don't see anything in the interferograms. Because of course the InSAR is just one
strand of all the information they have and all the contextual information they have about the community. So even if you might
be scientifically correct, your interpretation is of one dataset and you should never put it out there, you should never suggest
an action when there is someone who's job that actually is. You have to support them and not make it harder. So I personally
never tweet during a crisis. But I have quite often provided figures for volcano observatories to use. I think that's the right way
round".

To explore further the most impactful methods of geohazard research dissemination, we asked focus Group 1: "How should
scientists wanting to make the most impact with their geohazard research disseminate their results?". One academic participant
said: "If you rely on journals you reach your peers, a sub-section of media and a further sub-section of the public. Some of the
impacts we are talking about are relating to people on the ground affected by hazards, at risk of hazards. In those cases, you
cannot rely on journals to have these impacts. You must engage with people on the ground. Whether that's directly yourself
with communities or better through local agencies who have those partnerships". The group picked up on this reference of
communicating directly with local communities and many disagreed with this suggestion. Four participants emphasised, as
many interview participants did too, that communicating only with local partners should be the priority and not directly with
the public. One participant employed in a GRO said: "We have been looking at this project in [a country] and this engagement
of communities is actually very, very difficult. You have to do it really at a very high level because here there are just so many
people. But you can do it with different people. You can work with civil defense. You can also work with NGOs and this can
have a real impact. They have connections on much more local levels which can help you get engagement". Another group
participant also employed in a GRO agreed and said: "These things should be framed by local partners relative to the priorities
and social political sensitivities...everything that is going on in the local environment that we may not have full understanding
of. It has to be in association with user needs and defined really early on in the inception of a project". The group agreed on the
whole that, in a geohazard context, providing a range of uncertainties about future situations is more important than providing a
single future event forecast that may happen. One academic participant suggested expert elicitation is a good method to develop
these uncertainties. Other participants suggested simply translating research outputs into study region specific languages is





helpful for openly published results, as well as making data available in many file formats. Finally, one participant said: "When
you publish, maybe there should be additional effort to make clear the implications in our papers. Why not? Maybe it could be
helpful for those trying to synthesise the information, what are the clear results and clear implications".

     Many of our academic participants reported a lack of feedback on data and results shared online or with partners. This lack
of feedback is in contrast to, for example, the experience of those employed in GROs who aim to co-develop research based
on interaction with in-country partners over decadal timescales. One academic working on the Tomorrow's Cities project, for
example, described how in-country partners are built into projects from their inception. This participant's role was to process
topographic data and build flood models. In-country partners could not build such models due to limited compute power at their
institutes. This participant's research was published in scientific papers as well as fed-back to the Tomorrow's Cities project.
The participant, however, was not aware of how results were communicated eventually to local leaders, explaining they were
part of an established framework or "chain" aimed at reaching the end goal. They said: "The further you get to the end goal,
the more you lose site of what happens. Often you don't get feedback of how data is used unless you chase to find out, but this
is time-consuming. There is good feedback down the chain but less up it". Another participant described collaborating with
scientists in a country in Central Asia for 18 months to train them how to run seismic hazard scripts developed in the UK. The
hope was for these scientists to implement these approaches when they returned to their home institute. When asked if they
felt this project was successful, the participant said: "I think it was successful in that they had taken up some of the work. They
enjoyed what they did. I haven't then followed up to see what they are using, what they learned here, what approaches are they
using and implementing. I don't know the true final success".

     Finally, one participant who moved from working in an academic setting to an industrial setting explained how they needed
to significantly adapt how they explain their research to industrial clients as opposed to fellow academic researchers. They said:
"you get used to...in academia you either talk to people who use the same method and if you're not, you're still talking to a
researcher. I had to explain to someone what error bars were recently. They went "what is the vertical line? I don't know that
that is". They are not familiar with the concept of an error bar. You are not dealing with a scientist."

### 3.5   Openly available data

We discussed with interviewees the nature of openly available data, how reliant they are on this resource, and how useful this
open data is to the global community. We additionally asked focus Groups 1 and 2: "Should remotely sensed data and related
products (that focus on geohazards) be openly accessible for all?".

     Two participants who's academic work focused on constructing continental scale surface velocity fields if their projects
would be possible without access to open-access data. One of these participants responded: "Certainly not to the degree at
all. When we used to work [before 2014] the geographical range, the timescales and amount of data was not a range you
would choose because you had to get special accesses [to satellite SAR data] which would include the European Space Agency
(ESA), which meant it was a barrier to the volume. It slowed the whole process down. Now it is more...the barrier is not the
open availability. The problem is on your end, how much data can you suck up. ESA will give you as much as you want, it is
just do you have the capacity to take it and analyse it, process it, and think about it. That is the rate limiting step". This step





change in geographic and temporal scale of research was enabled by Sentinel-1 data which has been published open access now for over ten years, with many regions of the world being imaged every six or twelve days. We explored this "barrier"
with another participant. We asked if a non-expert would be able to use InSAR data. They said that most people who were an expert in any way would be able to understand and "cope" with deformation measurements, which are the final derived results produced by InSAR time series analysis. However, this participant acknowledged that InSAR data are complicated to interpret, let alone process. They said InSAR processing is a bit of a "black art" due to the many choices involved in producing interferograms (an image representing ground deformation over time due to a change in radar phase) and time series results
from SAR raw data: "You can kind of black box it [use default processing chains] but you don't always get the best results". A final interview participant has noticed a change since 2014 in the ability of volcano observatory staff to process InSAR data themselves. They said: "With InSAR these days they [observatories] tend to process a lot of data in house themselves so it's more a matter of giving feedback on what they have got or helping with a bug or helping them interpret it differently...But back in 2014-16 I was processing the data". In a similar way, a participant at a GRO used open-access data in combination
with open-source software to "empower" in-country partners to process data themselves. This approach is only possible due to release of Sentinel satellite data by ESA.

We asked participants if the satellite data that they use is open-access. As many of our participants worked with satellite SAR data, many responses focused on the openness of these data. One participant said in relation to InSAR data specifically: "The original raw data we now work with is very much open access. But in its form it is not very useable until you have to derive
the value added products which we have the processing chain here [where the participant works]. There are varying degrees of open, there are intermediate products, the interferograms which are open, the velocity fields only currently become available if they are published alongside the journal in some sort of open repository, but that is not systematic".

One participant in Group 1 suggested that data should be made available with agreement with local authorities and that for some regions you do not want to conflict with these authorities. Another participant suggested that there is a timescale element
to this data release and that sometimes volcano observatories will not immediately openly release data whilst an eruption is ongoing. Finally, a third participant explained that the Spanish authorities have an open data policy but if there is an ongoing geohazard event that the authorities shut down their web pages. This participant said: "If there is something going on they shut down their web pages, which doesn't help because people get really nervous. But on the other hand, it's always the question do you make it open for amateur interpreters".
In Group 2, made several considerations for whether data should be openly available or not. Firstly, if someone or some company is making money from this data then it should not be open. However, the same participant suggested that if the data is useful and it impacts other people, then this data should be open. The group also made the distinction between data being made open and available as opposed to data being accessible. One participant said: "I think related to hazards the data should not only be made accessible online but be made accessible for the people affected by it. And that usually requires working with
other people because that's not something we as scientists are best suited for". Conversely, a participant responded that data can be wrongly interpreted by the public. For example scientists know, they said, that volcano inflation will not always lead to an eruption. The public may not know this. They said: "My understanding is there should be some proper outreach activities





associated with this open access dataset". Building on this, a final participant suggested that if there are local authorities working on ongoing geohazard event, scientific results from the UK, for example, should not be published without local authorities and
their advice.





## 4 Discussion

The following section places our results in the context of existing frameworks that exist in both geoethics and disaster risk reduction disciplines.

Geoethics is defined as dealing with the "ethical, social and cultural implications of geoscience knowledge, research, prac-
tice, education and communication, and with the social role and responsibility of geoscientists in conducting their activities" (Di Capua and Peppoloni, 2019). This definition as well as a theoretical framework for geoethical thinking has been developed in recent years from a movement of opinion among geoscientists who have felt an increasing need to reflect on ethical values underlying geoscience practice (Peppoloni et al., 2019). The resulting framework consists of reference values and general principles to guide geoscientists towards taking responsibility for the Earth system (Peppoloni et al., 2019).

Disaster Risk Reduction (DRR) is understood to mean the development and application of policies, strategies, and practices to reduce vulnerabilities and disaster risks throughout society (Twigg, 2015). The relatively recent *Disaster Studies Manifesto: Power, Prestige and Forgotten Values* (Gaillard et al., 2019) provides a new academic "opening" (Khan et al., 2022) to address power imbalances and inequity in DRR practices.

The placement of our results in comparison to these disciplines is mindful of the current context in which researchers who
use satellite data to study geohazards reside. For many such researchers, the potential for satellite data to impact geohazard science has changed immensely in the last 20 years. Volumes of open-access satellite data freely available online has grown exponentially over this period due to programmes such as the European Space Agency's Copernicus missions (Jutz and Milagro-Pérez, 2020). This growth in open-data volumes combined with the UK's world-leading ability to process and disseminate large volumes of satellite data means that researching a geohazard outside of the UK without a field campaign is
easier than ever before. The ability to meet science goals and therefore impact policy from a UK remote sensing researcher's perspective is thus becoming ever more tangible (Elliott et al., 2016; Cigna, 2018; Tomás and Li, 2017; Wright, 2024).

### 4.1 Open-access vs equity of geohazard satellite data

Since 2014, our participants explain that the European Space Agency's (ESA's) decision to publish open access vast volumes of satellite data in near-real time through the Copernicus programme (Jutz and Milagro-Pérez, 2020) resulted in a step change
in how geohazard remote sensing researchers conducted their research. This change involved an increased ability to research larger spatial (up to continental) and temporal (in 2025 ~10 years) ranges in individual studies. Temporal resolution of this data is also more consistent and fine compared to previous satellite programmes and missions such as ESA's ENVISAT (Louet and Bruzzi, 1999) and ERS (Duchossois and Honvault, 1981) satellite missions.

However, participants and the wider community (Zebker, 2017) still identify barriers to accessing, processing, and interpret-
ing this vast satellite data catalogue. Many of our participants suggested that some researchers in the Global South historically or currently have reduced capacity to download and store this data compared to Global North colleagues. Additionally, due to reduced training and compute power, these Global South colleagues may have reduced capability to analyse and process satellite data and derived products. For InSAR data in particular, there is a long processing chain for constructing interferograms





and InSAR time series from the raw data released by ESA. Interferograms and time series are usually the derived products
that geohazard researchers use to monitor and research geohazards. This processing chain involves many choices including
which software to use (Li et al., 2022), whether to construct interferograms and/or time series data yourself or use openly
available products (Lazecký et al., 2020; Costantini et al., 2021), and which processing parameters are most appropriate for
the region of interest. These choices will depend on your research budget and time, compute ability and capacity, as well as
the characteristics of the study region. Some researchers with reduced compute power and time may choose, according to our
participants, "black-box" processing approaches in which SAR data is processed using default parameters and open-source
software. These approaches may not provide the most accurate, noise-free results. This reduced quality would impact result's
usefulness, certainty, and ability to be interpreted correctly. Accurate interpretation of satellite data is critical particularly in the
event of an evolving geohazard related disaster or hazard forecasting (Tralli et al., 2005). Furthermore, these barriers identified
by our participants imply a lack of equity in accessing and processing InSAR data to monitor geohazards. Equity can be defined
as existing when "everyone is able to achieve equal outcomes through equal access to opportunities and resources and to foster
a clear sense of belonging once those opportunities or resources are obtained" (Babinski, 2021; Nelson et al., 2022). From our
participants' experiences, a lack of remote sensing training, research time and budget, and overall research capacity limits the
equity of open-access satellite data for geohazard research.

There is, however, evidence from our participants for recent improvements in equity among the global geohazard satellite
remote sensing community. For example, over the last ten years, one academic participant described how their relationship
with a volcano observatory has evolved from them processing satellite data and providing results to observatories, to now ob-
servatory staff themselves processing this data. The participant now supports observatories more with providing feedback and
interpretation. This relationship evolution conforms with ambitions of the Disaster Studies Manifesto (Gaillard et al., 2019),
which states local researchers should be encouraged to lead research proposal development and design. This change may relate
to the increased volume of open-access data for volcano observatories to use. Additionally, this participant described how their
ongoing relationship with this observatory for nearly ten years has lead to co-supervision of PhD students and other co-learning
practices with the observatory to facilitate their increased research independence. Participants at GROs described similar ex-
periences in co-developing research projects with the overall goal not only of conducting research but to train and empower
in-country scientists. Sometimes, such international efforts are difficult to maintain for UK-based researchers however. One
academic participant, for example, described involving Central Asian scientists in an 18-month project to teach them how to
use scientific methods developed in the UK. The final success of this project remains unclear as the UK researcher lacks time
and funding to follow up on these collaborations. We see ethical review as a tool to better structure geohazard remote sensing
projects from their inception to include meaningful collaboration that meets the desire by geohazard researchers to involve
in-country researchers and ultimately contribute to DRR.

Further recent developments at the space agency level will further improve the equity of satellite data for geohazard research
at a global level. Firstly the European Ground Motion Service launched in 2022 serves open-access, land-surface motion data
for much of Europe (Solari et al., 2022). This service bypasses the need for InSAR compute power and expertise, improving
equitable access to InSAR derived ground motion data for Europe. Secondly, at a global level, the NASA-ISRO SAR (NISAR)





mission (Rosen et al., 2017) with a planned satellite launch data of March 2025 (Wang, 2024), has aimed from its inception to improve accessibility and usability of SAR products (Zebker, 2017). The NISAR team aims to deliver "ready to use" data without need for further processing, with published data using as small a storage volume as possible. Indeed, the NISAR team has conducted capacity-building efforts (Meyer et al., 2024), released sample data products ahead of launch (Zebker, 2017), and developed cloud-based training tools (Meyer et al., 2021). All these efforts will improve InSAR processing equity by reducing the compute burden for geohazard scientists, enabling those with limited storage, compute, or technological capacity an improved ability to measure and monitor geohazard deformation using satellite data.

## 4.2 Ethical reflection in satellite geohazard research

From our results, there is a clear misunderstanding from particularly the academic geohazard remote sensing community around the purpose and need to consider research ethics when conducting remote geohazard research. We find that remote sensing scientists, when asked about ethics relating to their research, often view ethical risks from a broad perspective and consider, for example, risks associated with personal privacy and high resolution imagery rather than ethical risks relating to their research methodology, aims, and desired impact. Moreover, an ethical review is viewed by some academics as an administrative checklist with black and white permissions that may completely deny researchers of their research plans.

Historically, Bond (2012) explains that the need for ethical review in research evolved from a need to avoid "atrocities" such as the Nuremberg trials in Nazi Germany (Kennedy and Grubb, 2000). These beginnings for ethical review have meant the principles underlying ethical review for many research disciplines still have their foundations in biomedical science and associated ethical risks. This medicine-centrist ethics system comes at the frustration of some social scientists who feel their work is unnecessarily upheld to the standards of medical experimentation (Schrag, 2011), with an "excessive fixation on harm avoidance" (Bond, 2012). However, we can reframe this focus on harm avoidance to better consider social and geophysical science practices. Bond (2012), for example, instead suggests ethical review should evaluate whether a planned study will achieve it's desired impact whilst maintaining the respect, rights, and well-being of human participants and others affected by the planned research. As well, researchers must take sufficient responsibility for the knowledge and findings that they present. These ethical considerations must be balanced against the "stubborn persistence of prioritising the quest for knowledge" (Bond, 2012) that some of our participants expressed. We found many academic research participants in particular considered their ultimate end goal of "saving lives" or "influencing policy" to ethically validate their study of any geohazard anywhere the world. This open perspective of our participants reflects the openness of vast satellite data repositories in 2023–2024. With no restriction to the availability of this data, remote sensing scientists perceive this lack of permission as positive permission to conduct geohazard research with no spatial restrictions. However, consideration should be taken to align the ethical considerations of geohazard remote sensing with those suggested by (Bond, 2012). Geohazard researchers should reflect if their study will achieve it's desired impact and if their research will affect the well-being of researchers and, in particular, imaged communities, even if these communities do not participate directly in the planned research.

Geoethical frameworks highlight "prevention" as one of the fundamental values underpinning geoethical practice (Peppoloni et al., 2019). Prevention refers to activities and tools which prevent processes happening or prevent resulting harm.



In the context of geohazard remote sensing, this value includes our participants desires' to "save lives" and "impact policy", whilst also guiding DRR practice. DRR focuses on preventing and mitigating against the impacts of geohazards on vulnerable communities. The Sendai Framework for Disaster Risk Reduction (Sendai Framework, UNWCDRR, 2015) is one framework which guides DRR principles and practice globally, aiming to strengthen DRR to reduce loss of lives and assets from disasters worldwide, 2015–2030. The Framework outlines four 'Priorities for Action' to inspire focused action on contributing to DRR, with the first being 'Understanding Disaster Risk'. To achieve this priority, enhanced access and sharing of 'remotely-sensed earth and climate observations' are specifically mentioned by the framework. Furthermore, academia is called upon to focus their research on disaster risk factors and scenarios in the medium and long term, as well as to support the interface between policy and science decision making. If communicated and disseminated effectively with policy and science decision makers as proposed by the Sendai Framework, research investigating geohazards is more likely to contribute to risk reduction and thus enhance community well-being, and ultimately save lives and influence policy. If these aims are achieved, geohazard remote sensing research will more closely align with geoethics and existing DRR frameworks. Conversely, poorly communicated and disseminated research may have the capacity for DRR due to it's scientific excellence but, if these findings never reach relevant decision makers or this research is not understandable and therefore useable by this community, this research will not be useful for DRR or therefore support the Sendai Framework.

However, there is strong feedback from our participants around the difficulties of forming and maintaining in-country collaborations, particularly in the context of academic culture, which inhibit academic contribution to DRR. For example, three interview participants mentioned that at least ten years of collaboration is required to make impact in relation to research influencing policy or changing societal attitudes, which are the desired impacts of many of our participants. This ten year timeline is in tension with the lifetime of research grants under which which many of our participants operate. For example, our participant who worked with seismology colleagues in Central Asia now lacks funding and thus time to continue their project development outside of the initial 18-month collaboration time-frame. In addition, there is academic pressure, as another participant explains, to focus on world-leading research with general focus rather than work with local decision makers on, for example, volcano monitoring: "I think if you stand up...and talk about monitoring at a conference people would take it a little bit less seriously. I don't want to say that because it is not uniformly true of course and lots of people are really interested in this area. I think the way that research is assessed particularly in the UK relative to the REF [(Research Excellence Framework, e.g. Torrance, 2020)] exercise, there is a lot of push to say "what I'm doing is globally applicable, what I am doing is uniformly useful, it's evidence of a big scientific process that's the same in all these volcanoes". And if you're working with a volcano observatory partner, they really care about what specific volcano you are working on right now, so in terms of preventing research and developing research that pulls us in both directions. So I wouldn't say that people are overly negative, I don't want to say people are scathing about it, but dismissive is probably a better word actually. I think there may be an instinct to be more dismissive about more applied things". This academic motivation to conduct high-impact work and seek prestige whilst overriding collaborative disaster studies with local researchers is noted as a concern by the Disaster Studies Manifesto (Gaillard et al., 2019). To better enable geohazard remote sensing scientists to contribute to DRR frameworks and have more



choice when considering ethical dilemmas, better education among the wider geohazard remote sensing community is required to highlight the benefits of establishing meaningful collaborations with the wider DRR community.

### 4.3 Experiences of collaboration

One critical ethical dilemma for geohazard remote sensing researchers within academia, therefore, is the need to balance conducting world-leading research to satisfy academic pressures with the responsibility that researchers hold for ethically disseminating and communicating their results. This dilemma is exemplified by the remote sensing geohazard scientist who published an article in a scientific journal highlighting elevated strain rates on a fault two years prior to a devastating earthquake. The researcher felt eternal guilt for not better communicating the potential risk to in-country people these strain rates indicated.

Additionally, contributing to DRR is difficult. Where efforts were made by our participants to better pin-point effective research dissemination to target in-country researchers or policymakers, some academics remained unsure how or if their disseminated research did impact communities or policymakers due to a lack of feedback back up the "chain" of communication to the researcher. We do find that some researchers, particularly working in volcanology, were able to engage effectively with local, in-country actors due to the establishment of volcanology observatories and associated worldwide volcanology networks.

Several volcanologist participants could point to specific examples of how there research or expertise influenced, for example, the boundary of an exclusion zone or the position of geophysical instrumentation. This influence contrasts significantly with that of earthquake scientists who found it difficult to engage with local decision makers when the probability of an earthquake rupture is so low in the short-term that earthquake risk is a low priority for policymakers, despite the potential damage and cost to communities being so high.

Achieving long-term in-country engagement, however, was more successful for those participants working within GROs. In these organisations, local partnership and engagement was common practice with participants aware of the complexity these relations can entail, especially when working with complex societal structure. Additionally, the approach to collaboration by GROs was different to that by academics. GRO participants focused on little and often engagement with in-country partners over many years to establish personal connections. Collaboration focused on co-development and co-production with GRO

scientists working towards in-country partners establishing research independence. Research outputs from this co-produced research, these researchers know, are more useful, useable, and used by societies vulnerable to the researched hazards. Such practices are key in decolonising knowledge production (Khan et al., 2022). (Khan et al., 2022) define decolonisation of knowledge: "An active, conscious and deliberate practice in the pursuit of truth and knowledge, which recognises the plurality and/or diversity of knowledge bases across the world; with an emphasis on knowledge rooted in grassroots communities, that

is effectively integrated in both the acquisition and production of knowledge". In a disaster context, disasters mostly take place in Global South communities, whilst representation from people living in these communities in western academic literature is minimal. In relation to satellite data, this trend relates to the inequity of satellite data accessibility and processing capacity, with the dominant direction of satellite data knowledge transfer from Global North to South. However, Global South knowledge has been valuable in, for example, local adaptation strategies emulated in the Global North formed by the lived realities of local

communities affected by disasters in the Global South. We recommend geohazard remote sensing researchers receive training





exploring neocolonialism in disaster research (Gomez and Hart, 2013) and how hazard and disaster research in the Global North can exert "soft power" (Andrabi, 2022) towards Global South communities. Training should suggest measures that researchers can take to decolonise their research methodologies where appropriate by engaging with local partners and disseminating research outputs in an accessible way for in-country end-users. It is important this training is specific and actionable to remote sensing researchers and considers the complexities surrounding research decolonisation practices (Barnes, 2018).

This contrast in experiences by academic and GRO participants demonstrates contrasting alignment of these two groups' with the Sendai Framework. The Framework recognises that, despite the remote sensing community having a role in producing and communicating research to understand disaster risk, states have the overall responsibility for reducing disaster risk(UNWCDRR, 2015). This DRR responsibility is not up to the individual researcher. Each actor in a disaster and hazard context has specific roles and responsibilities and is part of the "defence system" against potential risk (Peppoloni et al., 2019). This need for researchers to engage with states and relevant decision makers is exemplified by the ESPREssO project. In 2020, the ESPREssO project (Zuccaro et al., 2020) aimed at a European level to identify gaps and priorities for not only DRR but also Climate Change Adaptation, and Disaster Risk Management within the Sendai Framework. They found that despite the "overwhelming amount of information available through satellites, remote sensing, public repositories, social media, data are scattered among multiple sources, thus hampering their effective use and translation into actionable information for decision makers to support their activities in the whole emergency cycle". Additionally, ESPREssO highlighted among decision makers and other end-users their ability to understand high-level information from these scattered sources. Importantly, COMET themselves acknowledged the potential for their datasets to support DRR, but find that their communication strategies through, for example, Twitter (now X) are more oriented to disaster and event response rather than towards hazard preparedness or DRR (Watson et al., 2023). It is therefore critical that geohazard remote sensing scientists understand and act on their role in the Sendai and other international frameworks to enhance data dissemination and reduce academic neocolonialism. Remote sensing researchers have an indispensable role in working with states to communicate effectively their findings to align their research strategy with geoethical principles to DRR. Such understanding and action will prevent overwhelm by remote sensing scientists who feel they lack sufficient training and knowledge to contribute to DRR and "saving lives", despite their best ethical intentions.

### 4.4 Post-disaster publication gold rush- a neocolonialist practice?

Academic participants instead focused a lot of attention on collaboration during a funded, time-limited projects or during specific disaster events. For example, focus Group 1 identified hypothetical and real examples where academic researchers "rush" to study a rare, unique event perhaps not observed during the modern satellite era. These participants explain that if enough data is available then in-country collaboration isn't needed and that "work did not wait" for external researchers to reach out to in-country scientists, highlighting the perceived priority of research excellence over local collaboration. This contact with in-country scientists as simply a need to request permission rather than as an invaluable tool to interpret and understand an evolving hazard in a disaster setting. Such "gold rushes" for scientific publications and world-leading research in the immediate wake of disasters are well documented. Gomez and Hart (2013), for example, note that the number of articles





on Scopus featuring "tsunami" increased 10-fold between 2004 and 2005 following the Sumatra-Andaman earthquake and tsunami disaster of December 2004 (e.g. Fujii and Satake, 2007). The authors argue this voluminous post-event literature is often neither of high quality nor contribution to scientific progression. Fisher et al. (2021) reflects on a similar phenomenon for archaeologists who conduct remote sensing studies. They similarly explain that remote sensing differs significantly from physical fieldwork because the former does not require engagement with local regulatory frameworks. This lack of engagement

creates a significant shift in power towards the external researcher, they explain, and places with them full control over data and narratives, bringing risks of "digital colonialism". Such lack of engagement in local frameworks and communities results in such research being uninformed by local realities, a concern noted by the Disaster Studies Manifesto. This lack of engagement means the risk that people experience from geohazards is not incorporated into geohazard research (Gaillard et al., 2019).

This rush for research success is not total among our participants' experiences, however. One participant explained in some

instances academic researchers take care in "staying out of the way" of catastrophic disasters and instead try to engage with in-country researchers remotely to share results. This behaviour reflects a desire among researchers to make the best ethical decisions both in general and during geohazard related disasters. We additionally felt a desire among academic researchers to better contribute to DRR efforts and projects. These researchers, however, felt that they had a lack of experience doing so, saying that they lack necessary training, time, and funding in purely geophysical research projects to take their results

further and disseminate material which is useful, useable, and used. There are now stipulations by some journals for open-access release of code and data relating to such geophysical projects, for example, but this release is often targeted towards fellow scientists rather than towards decision makers or DRR specialists. Additionally, many academics do not receive or seek feedback on their dissemination or collaboration efforts due to a lack of time and funding. One participant described the desire to make sustainable impact desired by researchers (such as impacting policy) is in tension with both university

research demands for world-leading research, and the wider research community's sometimes dismissive attitude towards, for example, volcano monitoring. Such tension can push geohazard remote sensing researchers towards conducting fully remote research without engaging in-country partners, leading to more academic neocolonialist practices (Gomez and Hart, 2013). These practices are at odds with academics' ethical desires to make a meaningful difference to communities vulnerable to geohazards.

Finally, a rush to publish satellite data derived results relating to geohazard research may come with risks for the geohazard researcher. In April 2009, an earthquake struck the city of L'Aquila in central Italy, killing over 300 people and injuring 1,500 (Alexander, 2010; Di Capua, 2013). In the wake of this disaster, six scientific members of the Major Risks Committee of the Italian Government and a researcher at Istituto Nazionale di Geofisica e Vulcanologia (INGV) were found guilty of contributing to many deaths due to their inappropriate communication of appropriate seismic hazard and vulnerability (Di Capua, 2013).

Geoethical reflections on these events cite a need for improved quality of scientific communication by geohazard scientists, using audience appropriate language that is still scientifically correct, as well as a commitment to conduct reliable, updated scientific research with uncertainty and error evaluation (Di Capua, 2013). Furthermore, relationships with media and policy makers must consist of sharing operational protocols and establishing defined roles and areas of expertise. There is evidence from our participants of their strong awareness to perform only their scientific responsibility in providing facts and results





relating to hazard characterisation before, during and after a disaster. This practice conforms strongly with geoethical recommendations (Peppoloni et al., 2019). Participants recognise these results should be fed through established channels directly to local decision makers, and researchers should not, for example, provide opinions to media on, for example, when an earthquake will rupture. These behaviours by our participants are generally learned informally between colleagues and through observation of events such as those in L'Aquila. Formal training around geohazard scientist involvement in ongoing disasters, as well

as around scientific publications in the "gold rush" wake of disasters, should be implemented to formalise these practices.

Overall, researchers should consider the real tension in the "gold rush" wake of a disaster that exists between the need to capture essential data for future disaster prevention and the need for collaboration in these times. Scientists should consider the research opportunities and often practical necessity of in-country collaboration before rushing to retrieve post-event data.

### 4.5 Geoethical decisions are context dependent

As discussed by (Peppoloni et al., 2019), geoethics varies in space and time as social, political and economic contexts that inform these ethics evolve. Therefore, decisions and actions taken by geohazard researchers that are informed by geoethical principles may vary, even if the ethical dilemmas themselves remain constant. Researchers must therefore continually reflect, asking what is the appropriate ethical choice given the current context. This context dependent ethical thinking has historical precedence in relation to remote sensing. For example in World War I, planes carrying cameras frequently infringed on

national airspace as, up to this point, there had been no international agreements governing air space use by foreign states (Wasowski, 1991). This changed in 1919 with the Paris Convention which stated a nation's sovereignty exists above all territory which it controls. It was not until during World War II when the military importance of aerial imagery was demonstrated, and consequently the 1944 Convention of International Civil Aviation granted contracted states the ability to prohibit operation of imaging devices within their airspaces. This example demonstrates how society has been historically reactive rather than

proactive regarding what it will legally and therefore ethically accept in relation to remote observation. Extending this logic to the present day and our study, many participants explained how they did not consider people or society imaged in their research to be an ethical issue due to the resolution of the data that they use being too coarse. However, satellite data spatial resolution is in constant improvement due to evolving global technological capabilities and thus geohazard researchers should reconsider these ethical viewpoints as their datasets develop. For example, at the time of writing, Wasowski (1991) and the wider pub-

lic could only speculate on the technological capabilities of the US satellite photographic reconnaissance programme, which was acknowledged by President Carter in 1978 (Richelson, 1984). Precise knowledge of the sheer geographical scale and fine spatial resolutions at which these US spy satellites under the CORONA programme (1958-1972 Cloud, 2001) were acquiring was not apparent until the declassification and release in 1995 of CORONA acquired optical imagery. The KH-4B satellites (1967–1972) of the CORONA programme acquired images with spatial resolutions of up to 1.8 m (Goossens et al., 2001), far

superior to any publicly available satellite imagery at the time (80 m, 1972–1982, Landsat 1–3; 30 m from 1982 with Landsat-4 Williams et al., 2006; Goward et al., 2006). It was not until 2000 (IKONOS-2, 1 m resolution) when satellite data became commercially available at spatial resolutions comparable to those possible since 1967 under the CORONA programme.



In relation to our study participants, this story of the development of military satellite capabilities away from academic and public applications may have future parallels. There is currently minimal concern among our participants for the ethical implications of their work, which is mostly conducted at tens of metres spatial resolution. However, it is difficult to predict how data available to scientists may advance given that some technologies may remain unknown, being classified for example for military purposes. One technology in development, Synthetic Aperture Laser (SLA), is capable of imaging motion and deformation at spatial resolutions much finer than the current commercially possible 1 m limit (Turbide et al., 2017). If such satellite data was available for use by geohazard scientists, they may be able to track the movement of people, vehicles, and other objects. This advancement is in contrast to the data that most of our participants usually use, which has spatial resolutions in the tens of metres and thus, many participants identify, is usually unable to image people or other individual identifiers. One participant from industry does acknowledge that as very high spatial resolution data (<1 m) is becoming more accessible, more of their colleagues in industry are suggesting greater care is taken to mitigate against potential ethical risks. In a rapidly developing technological field, geohazard remote sensing scientists should consider the dynamic nature of their available datasets and thus, in accordance with geoethical thinking, the need to revisit ethical risks as new data and contexts emerge. Remote sensing geohazard scientists should therefore be prepared to modify their publication and dissemination strategies in the advent of new types of data becoming available. Universities and research organisations should offer support and guidance on considering ethical dilemmas associated with new technologies. This guidance should support individual scientists in exploring the strengths and advantages that increasing spatial resolutions and other technological developments can bring to disaster science, as well as potential ethical risks. Such guidance would allow these scientists to make informed decisions regarding satellite data use and dissemination.

The history of US satellite image resolution and data dissemination demonstrates the control that satellite data image providers have in releasing information. Still today, satellite data acquired at ∼1 m resolution or less is only available for military use, governmental use, through commercial agreement (Dumery et al., 2018), or through data agreements related to scientific goals such as the Committee for Earth Observation Satellites (CEOS Lowenstern et al., 2022). This control allows image providers to decide if imagery is appropriate to release to, for example, academics for research purposes. From our results and published literature, it appears this control relates mainly to the timing of imagery release. For example, the decision by the US government to declassify high-resolution CORONA imagery— which captured politically sensitive events and sites such as the Vietnam War—- decades after acquisition removes an ethical risk and consideration for imagery end-users. End-users and researchers do not need to consider if the imagery that they are using captures events or information that may impact international relations or ongoing conflicts. Indeed, in recent years, scientists have used this declassified satellite imagery from CORONA and another US spy satellite programme KH-9 HEXAGON to locate bomb craters from the Vietnam War period (Munteanu et al., 2024; Barthelme et al., 2024). The KH-9 HEXAGON archive was fully declassified in and released in 2011 (Dehecq et al., 2020). This time delay was mentioned specifically by one of our participants as a measure that could be taken by researchers to mitigate ethical issues around publishing satellite data results in a region of ongoing conflict. They suggested a time lag of, for example, five years after acquisition for publishing such results may mitigate published work being perceived as interfering with ongoing conflict. For more modern imagery, one of our participants explained that commercial satellite data





providers sometimes limit completely the release of very high resolution satellite data (such as TerraSAR-X, spatial resolution up to 1 m) which image regions in conflict such as the Ecuador-Colombia border, perhaps as this imagery was used for military
purposes. This restriction removes some need for end-users of imagery, such as geohazard scientists, to consider the ethics of using imagery in their research which captures conflict. Still, researchers who are provided with ∼ 1 m commercial imagery are unable to directly publish this imagery, but are able to publish results related to these images. Geohazard remote sensing scientists therefore have a personal choice in regards to result sharing- the individual researcher must consider any ethical risks that they foresee in publishing related results, particularly if these results will be published open-access.

Aside from resolution, political and societal attitudes influence which ethical risks are perceived to be acceptable in the context of geohazard research. Several participants with longer remote sensing careers noted a change in attitudes towards the ethics of "parachute science" and the involvement of local actors when conducting remotely studying geohazards using satellite data. One participant noted how now compared to twenty years ago researchers in ex-colonial nations such as the UK are more aware that conducting research in another country, particularly if that country was previously a colony such as India to the
UK, would appear colonial in nature. This participant's view highlights how geohazard researchers' attitudes about remotely conducting research have changed over time, perhaps being shaped by changing societal views on colonialism and a growing awareness of the need to decolonise research practices (Khan et al., 2022).

### 4.6 Consent of imaged societies

Another ethical consideration related to remote observation by satellites from space relates to consent by imaged societies.
Considering another historical example, after the first non-military satellites were launched by the Soviet Union and the United States in 1957–58, there were no notable objections that their orbit may impinge on national airspaces (Wasowski, 1991). After a prolonged period of unchallenged satellite flight, it became *customary international law* that any nation had the right to overfly another by satellite without seeking permission. However, some claimed that they might have "strenuously" objected had they been better informed (Hingorani, 1988). This idea of remote sensing observation without positive consent relates directly to
themes that emerged in our study. All interview participants said that people and society imaged by satellite data at whatever resolution had no say in whether this imagery was acquired. Moreover, in general, participants in FGDs and interviews did not see the need to take precautions when sharing results derived from satellite data as imagery captures generally just infrastructure rather than individuals. One participant did identify that interpretations of satellite data are more "ethically problematic" than the data itself. This opinion contrasts that of other participants who claim the data they use is not dissimilar, even at high
resolutions, to that available on Google Earth. Furthermore, another participant highlighted that in some cases, providers such as Google Earth do in fact take measures to pixelate some regions at war. We argue that the added interpretation and analysis added by geohazard researchers to satellite data give the geohazard researcher great ethical responsibility towards people and societies captured in their data. For example, publication of a hazard map may influence house prices, as has been observed by one participant in the Azores. Furthermore, these communities are most likely captured without consent or perhaps even
without knowledge of the capabilities of satellite imaging systems to image their lives. This lack of consent and awareness should be considered by geohazard researchers in the context of potential ethical risks. These specific considerations develop





over time with the evolution of satellite data available to researchers, publically or commercially, requiring researchers to react dynamically to evolving ethical risks.

## 4.7    The need for remote geohazard research and monitoring

On the whole there was recognition by our participants in academic, industry, and GROs that long-term engagement with in-country scientists and decision makers would improve the quality and relevance of research to communities, as well as improve research impact. This recognition is aligns with the recommendations of the Sendai Framework.

However, as highlighted by our participants, in some contexts there is a real need for remotely studying geohazards without in-country partners engagement. On a practical level, for example, some mountainous regions of the world have extreme
terrain and climate which puts in-field researchers at risk. Additionally, as there is significant volume of openly available satellite remote sensing data, this data can be used to mitigate against researchers may encounter in, for example, accessing study sites during a global pandemic.

Sometimes remote sensing studies have value in reducing disaster risk even if collaboration with in-country partners is not possible. This is true for countries and societies experience difficulties in establishing collaborations externally, which can
limit the quality of research studying geohazards in these regions. Sometimes in-country scientists resources' such as time and compute resource are stretched, for example, meaning they may be unable to invest time, money or compute time in international collaboration. These scientists may welcome support from external scientists in contributing to in-country DRR. Conversely, political limitations may impact in-country scientists' ability to collaborate. For example, research conducted outside of Iran using satellite data without engaging with local partners is welcomed by Iranian scientists due to the impact
of sanctions on the ability of Iranian scientists to collaborate with external scientists. In these examples, researchers should consider how working without in-country partners might impact the quality and impact of their work, and how these impacts can be mitigated to the best of their ability. For example, perhaps collaborations can be made with Iranian researchers who work outside of Iran. These external Iranian researchers may be better informed of the most appropriate and effective communication channels, such as social media, to disseminate geohazard science back to the country.

## 5    Future Recommendations

To achieve ethical behaviour in a community, ethical behaviour should be firstly accepted as an excepted norm; ethical behaviour should secondly be taught and modelled in both formal and informal educational settings; and thirdly unethical behaviour should be identified as unacceptable (Bobrowsky et al., 2017; Peppoloni et al., 2019). In this study, we see evidence that some geohazard remote sensing researchers exhibit geoethical practice, particularly in relation to conducting high quality
research and some awareness and ability to engage with local decision makers. This evidence suggests there is some satisfaction of this first criterion: ethical behaviour is accepted as an expected norm. However, to enhance community awareness of the need for practicing ethical geohazard remote sensing research, further geoethical training and education is required to fully achieve these first two criteria. However, we recommend geohazard remote sensing researchers receive training to promote the



need and ability to reduce ethical risks in their research. This training should be grounded in geoethics and existing DRR prac-
tices, as well as the principles of the Disaster Studies Manifesto (Gaillard et al., 2019). Throughout, training should emphasise
that consideration of ethical risks during study design periods rarely results in research being denied and completely cancelled.
Instead, researchers should understand that better awareness and consideration of ethical risks allows for study designs to be
altered and improved to ensure that ethical risks are mitigated in the study design process.

We recommend training should focus five main topics: a) best practice to enable decolonisation of geohazard research; b) best
practices remote sensing research for aligning with the Sendai Framework and other international DRR agreements; c) effective
research collaboration and dissemination strategies whilst mitigating ethical risks involved in international collaboration; d)
examples where in-country collaboration is not appropriate and practices for dealing with these situations; and finally e)
strategies to protect imaged communities as earth observation technologies evolve. In addition, we propose a light-touch ethical
review system at the project design stage to allow ethical risks to be identified and mitigated against. We expand briefly upon
these five training topics below.

Geohazard remote sensing researchers should receive bespoke and specific training that explains the need for decolonising
research and knowledge creation in their field, as explained in this study. This training should provide guidance on practices
to reduce colonialist aspects of their research strategies. These practices are better established in adjacent disciplines including
DRR through their strong adherence, for example, to the Sendai Framework. This training should therefore take precedence
from established DRR practices in establishing in-country partnerships where appropriate, and how to assess and mitigate other
ethical risks.

Geohazard remote sensing scientists must receive specific training and guidance on how their results and findings are es-
sential in aligning to international DRR frameworks as well as, for example, the Sustainable Development Goals (Hák et al.,
2016). Building on this knowledge, best practices for integrating these results and findings into established DRR frameworks
and workflows should be demonstrated. These best practices may include collaborating with DRR specialists, such as social
scientists, at the research project design stage to ensure a more holistic view of DRR is employed throughout the planned
research beyond hazard characterisation.

We therefore recommend that researchers involved in publishing geohazard research using satellite data consider, from
project inception, the most appropriate method to communicate their findings to protect the well-being of vulnerable commu-
nities. This recommendation may require a change in the approach to study design as well as the training offered to geohazard
remote sensing scientists around engaging in effective dissemination networks. Extensive frameworks and guidance already
exist to advise best DRR. Alongside this push for collaboration should be a discussion around ethical risks and power dynam-
ics encountered when communicating and disseminating results with local, national, and international audiences and partners.
These risks should be considered separately for potential hazards and ongoing disasters, given the differing urgency in each
case. Training should discuss the complexities of establishing and maintaining collaboration with in-country partners, empha-
sising the benefits of collaboration in terms of achieving impact, improving access to local data, and decolonising research
practices.




Exceptions to establishing collaboration should be discussed with examples drawn from situations similar to those experienced by our participants, with guidance on how best to communicate research when, as a last resort, local partnership is not possible. For example, dissemination of key datasets and conclusions online via open-access portals, journals, and through social media permits some potential quick and free communication of results to local stakeholders. However, the success of such dissemination pathways is often unclear. This training will enable researchers to critically reflect and improve research aims and study design to align with international DRR frameworks and contribute to the decolonisation of remote sensing research. Such training aligns with the geoethical principles.

In an ever advancing technological world, remote sensing technologies are no exception. With future satellite instrument technological capabilities at best speculative, remote sensing researchers must be prepared to reflect on ethical risks involved in emerging technologies that are released. It may be difficult to foresee these new technologies as some may be deeply classified as in the case of the US reconnaissance satellites. Training should focus on strategies researchers can employ to protect themselves and imaged communities when using these future technologies..

Finally we consider the third action required to achieve ethical action in the national and international geohazard remote sensing community: unethical practice should not lead to research advantages (Peppoloni et al., 2019). In practical terms, we suggest that this could relate to grant and research funders stipulating a consideration for collaboration or co-development of research with local partners, strong plans for ethical research dissemination, or evidence of how remote sensing research will support the Sendai Framework or SDGs. Such stipulations are more common place in, for example, DRR research communities. Such action could begin to influence the geohazard community more broadly and address concerns raised by participants in balancing funding, university, and community needs for academic excellence with a desire to contribute to DRR. Furthermore, we don't necessarily recommend traditional ethical review for all geohazard remote sensing studies. However, we believe a light-touch ethical review system of geohazard remote sensing research could be incredibly valuable. Such a system could aim to highlight unforeseen ethical risks in study designs including lack of thought for meaningful result dissemination, or allow ethical review panels to suggest measures to reduce academic neocolonialism. In addition, reviews could suggest improvements to research plans and study designs to increase engagement and alignment with, for example, the Sendai Framework. Finally, the review system could consider and provide feedback on ethical risks that arise with advancements in remote sensing technologies, such as finer spatial resolutions. These top-down encouragements to consider ethical risks in geohazard remote sensing research would support bottom-up ethical training of individual researchers.



## 6 Ethical Statement

Participation in all interviews and Focus Group Discussions in this study was voluntary. Participants were informed before their participation that they were able to withdraw up until dissemination of study results. Data were pseudonymised upon collection. Participants completed and returned to the study coordinators a study-specific Consent Form. The University of Leeds' Business, Environment, Social Sciences joint Faculties Research Ethics Committee granted ethical review for this study.

*Data availability.* Due to the potential sensitivity of interview and FGD material, as well as the possibility of interview and FGD material being used to identify participants, we choose not to openly release resulting manuscripts or recordings with our manuscript.



**Appendix A: Supplementary Methodology**

**A1    Participant Sampling**

**A1.1    Interviews**

Potential interview participants were identified initially by personal contacts of the authors. These participants were sent an invitation by email that included a Participant Information Form and Consent Form designed for this study. This invitation made clear that participation was non-compulsory, with participants given time to decide privately if they wish to participate in the study. If these invitees wished to take part, we provided 1-2 weeks for potential participants to consider their participation

and return these two forms.

The Consent Form provided options for participant anonymity, with participants having the option to keep all, none or some of their age, gender, nationality anonymous.

As one of the authors of this study was participants in the UK academic satellite remote sensing community, some of these COMET participants were personally known by the authors. Recruitment of these known colleagues therefore offered

an accessible source of research participants. However, these participants may have felt an obligation to take part due to being know by the authors. To reduce the risk of this unintended coercion, it was made known to participants at every stage of recruitment that participation was optional.

In total, we conducted thirteen interviews. This relatively small number of interviewees provided sufficient amounts of data to answer our study aims whilst also maintaining a manageable number of interviews to conduct and analyse. In this group

of interview participants, ten self-identified as male, three female; eight employed in academia, two in industry, two in both industry and academia, and one in a research organisation. Seven participants had used satellite data in their job or career for 10-20 years, two for 20-30 years, three 5-10 years, and one 1-5 years. Eleven participants were active members of COMET at the time of their interview. Two participants were professors, three postdoctoral research fellows, one associate professor, and the rest mostly identified as some kind of scientist including remote sensing scientist, data analyst, or software developer.

**A1.2    Focus Group Discussions**

COMET Annual Meeting participants were emailed two weeks in advance in advance of the planned focus group session an online consent form for participation in the FGDs. This form also collected participant information with the same self-identifying questions as for interview participants.

Including participants in all three groups, there were fifteen self-identified male and seven female participants; eight British,

six non-UK European, and six non-European participants; fifteen from academia, one working in both academia and industry, and all others working in a government or other research institute, agency, or organisation. There were seven postgraduate researchers (one of which also lectures), four professors, two lecturers, one postdoctoral fellow, and six who identifies as scientists including remote sensing scientists. One participant had worked with satellite data for less than a year, ten for 1-5



years, seven 10-20 years, and three 20-30 years. The breakdown of participants into the three focus groups is illustrated in Figure 1.






**Figure A1.** FGD participant demographics illustrated by bar charts. Each column represents a group of participants and each row describes a self-identified characteristic of participants in that group. A + I = Academia and industry; GRA = Government Research Agency; ARI = Applied Research Institute; RO = Research Organisation; RI = Research Institute; S. Academic = Senior Academic; PGR = postgraduate researcher; Postdoc = Postdoctoral Research Fellow; RS = Remote Sensing; NH = Natural Hazards. The final row 'Years' denotes the number of years the participant has used satellite data in their career or job. Note we have aggregated in Group 3 British and English participants into one category.





## A2 Data Collection

### A2.1 Focus Group Discussions

FGD Participants were provided with a brief verbal introduction to the research and how the FGD session would run, including an emphasis on participation being optional, followed by time to ask questions. Each group took their seat with their first
moderator and each participant introduced themselves to their group. The moderators then began the discussion by posing the first question for their topic. The FGD audio was recorded throughout by the moderators using a dictaphone or Microsoft Teams. Each group of participants rotated between moderators every ten minutes. Given the unforeseen time constraints, moderators did not always have time to ask all questions depending on the depth of initial discussion of each topic within each group.

FGD Moderators were staff or students at the University of Leeds in the authors' wider research group, who were selected based on previous experience running focus groups or research group discussions. A week in advance of the FGDs, moderators were provided with guidance to the FGD aims, how the FGDs fit into the wider study aims, guidance on moderator conduct, and a 'Topic Guide'. This week allowed time to digest and ask questions to the study leaders. The 'Topic Guide' set out the three key topics to be explored during FGDs, providing three questions per topic for moderators to ask. Moderators were guided
to intervene as little as possible, allowing discussion to be stimulated from the Topic Guide. Intervention was encouraged only to clarify participant meaning and stimulate discussion.

## Appendix B: Study Limitations

This study focused on capturing the opinions of UK-based remote sensing research scientists, 2023–2024, who study geohazards. These scientists were mainly part of COMET and related personal networks. This focus on COMET lead to a strong focus
in our collected data on InSAR and other SAR data products and their use, rather than on other satellite data products such as optical, altimeter, or thermal data. Further work could aim to better sample researchers who use these other datasets to study geohazards. Additionally, similar studies could be conducted among geohazard remote sensing communities in other countries to see how ethical opinions vary.

We acknowledge that our findings represent a small snapshot in time of geohazard remote sensing perceptions and opinions
from the perspective of researchers trained as data scientists, geophysicists, or remote sensing scientists working in a Global North research setting. As a demonstration of how communication strategies changed for these researchers during the study period and up to publication of this manuscript, the social network "Twitter" was bought and rebranded to "X" in July 2023. This change may have significantly affected researchers who relied on Twitter as a communication strategy. COMET themselves used Twitter (and currently rely on X) alongside open-access data portals to disseminate ground deformation data (Watson
et al., 2023). Twitter before July 2023 was also used as a platform for geoscience discussion and collation of geohazard events by the wider community (Lacassin et al., 2020; Ruan et al., 2022). Due to the timing of Twitter's transformation towards the end of most of our data collection (after eleven interviews and FGDs), we could not fully explore it's impact on geohazard remote



sensing communication and result dissemination. Future work should aim to explore the impact of this and other changes in social media platforms on geohazard research dissemination strategies and success.



*Author contributions.* JAP and KD conceptualised the study aims and methodology. JAP planned interview and FGD questions, and their formats. JAP performed interviews and oversaw moderators during FGDs. JAP performed data analysis and wrote the article original draft. KD supervised and reviewed all stages of study design, execution, analysis, and writing.

*Competing interests.* The authors declare that they have no conflict of interest.

*Acknowledgements.* The authors wish to thank Ruth Sylvester for her support in the ethical review process. We additionally acknowledge
FGD moderators Qi Ou, C. Scott Watson, and Eilish O'Grady. Finally, we thank COMET managers for allowing us to run a session at their annual meeting, and COMET members and all other study participants for their honest insight into the world of geohazard satellite remote sensing in the UK.





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
