# Peer review of "The ethics of using satellite data to monitor and publish research on geohazards in regions of political complexity"

_EGUsphere, 2025_

## Referee Comment (RC1)

**Review of: "*The ethics of using satellite data to monitor and publish research on geohazards in regions of political complexity*" by Payne and Donovan, submitted to Geoscience Communication.**

**Reviewer:** Dr Sam Wimpenny, University of Bristol (UK)

**Note:** In full disclosure, I am a member of the group from which the remote-sensing scientists was solicited, though I was not part of this study in any way. I am also not an expert in ethics. I would recommend that any other reviewer of this article is someone with expertise in science ethics. However, I do believe I can offer an independent review of the methods, the utility of this work to the remote-sensing community and highlight areas where interpretations/arguments could still be clarified.

I also apologise to the authors that it took my so long to write this review.

**Article Summary**

In this submission, the authors evaluate and review the ethics of using remote-sensing data in scientific research through the testimony of UK-based remote-sensing scientists. Their primary dataset is transcriptions of interviews, focus groups and break-out sessions, and they evaluate the responses of their target group within the framework of existing policy and history of remote sensing ethics. The authors conclude that, despite there being awareness amongst their sample of remote-sensing scientists about there being moral dilemmas when using or creating datasets from remote-sensing products, there is some reticence to introduce any formal procedure for reviewing practice and a lack of understanding of the range of ethical questions beyond those related to identifying individuals from satellite data. They highlight tensions between the pressures that researchers feel from policy and assessment criteria (e.g. REF) and established best practice frameworks.

Overall, I believe that the article highlights important limitations amongst remote-sensing practice in the UK – particularly that there is an underappreciation of the agency that researchers have to choose what to study, how to design their studies, and how their data is shared or distributed. Much of the testimony from researchers appears to be split into either 'good' versus 'bad' research practice and this highlights the bimodality in which many practitioners see ethical review, as opposed to an opportunity to retain the scientifically 'good' aspects of their work but ensure that the potential for 'bad' is minimised. The article may therefore also be important for advocating for changes to existing practice in reviewing research quality and impact both at an institutional level and through nationally significant programs like the Research Excellence Framework (REF).

One limitation of the article in its current form is that, although it outlines a series of useful actions to address some of the limitations of current remote-sensing practice, who is responsible for these actions (e.g. training on decolonial approaches, collating and establishing best practice, conducting ethical review) remains unclear. To ensure that the article changes ways of working and has the desired impact the authors should consider identifying who should enact their recommendations.

There are some further changes that I recommend the authors make to the manuscript to clarify arguments, significantly shorten the text (if possible), more clearly establish the scope of the study, and highlight the limitations more clearly. These constitute restructuring and clarification rather than technical/methodological problems. *I therefore recommend the article could be accepted following these minor revisions.*

**General Comments**

1. **Shorten Introduction and clarify the scope of the article:** I found that the proposed scope of the article laid out in the Introduction did not always align with the material that was drawn from the transcripts in Section 3, or within the discussion Section 4. The authors state that this article is about ethical dilemmas when studying geohazards in regions of 'political complexity' (a term that would benefit from being defined more clearly) and that it will focus on 'land subsidence in a region that is politically complex' (Line 29-30). My assumption was that the authors would then draw on moral questions that emerge from their discussions with practitioners when analysing remote-sensing data and reflect on their research context. The ensuing discussion is far more broad reaching, in terms of the types of geohazards (earthquakes, volcanic hazards, land use, population, urban mapping, gas measurements), political context, geographic scale (global vs state-wide vs local), and stakeholders (data distributors, practitioners, research end users, general public). Some of these topics even stray from what I understand to be ethical questions that research practitioners must consider in their project design (e.g. commercial data sharing policies). My recommendation would be that the authors consider restructuring their introduction to explicitly define what they consider to be 'ethical questions' and to more clearly define the scope and stakeholders considered in the article.

   Lines 61-120 draw heavily from Wasowski [1991] and existing policy documents. The authors could consider whether it is reasonable to cite this material and then draw from it more generally in the subsequent discussion, rather than have a description of their arguments within this article. This would help shorten the introduction and clarify the article's scope.

2. **Justification of the grounded theory approach and acknowledging limitations:** The methodology and motivation is clearly explained but there could be a clearer acknowledgement of the limitation of the sample size and the sample demographic within the main text, as well as a discussion of the implications of these limitations. The authors could also explain how they have considered these limitations when analysing the transcripts and selecting content from them within their methodological overview.

   For example, the testimony that forms these discussions comes from people who may not fully understand local ethical issues in the regions they are working, such is the nature of remote sensing. Are these people best placed to identify and challenge the ethics of their own practice in relation to local collaboration? This is highlighted in testimony from some participants, but it could be clearly outlined in the methodological section.

3. **Synthesis of the results in Section 3:** The authors should consider ways to synthesise or distil the results of the FDG and interviews more effectively – potentially include visuals. This could involve a set of tables of selected quotes and responses for specific questions, for example, with the detailed analysis of these responses remaining in the text. Alternatively, the authors could consider other visual ways of summarising responses (e.g. speech clouds). As a reader, I did find it hard to keep track of the contrasting sentiment between groups and summarise the responses across different questions, as the article is quite long.

4. **Summarising quotes:** This might be intentional on the part of the authors, but I did find some content of the quotes to be unnecessary and detract from the key message. Consider paraphrasing the quotes to remove unrelated content.

5. **"Education" as a solution to the conflict between local collaboration and academic pressures:** Section 4.2 highlights the challenge of creating meaningful, long-term partnerships with local researchers and how it can be in conflict with academic pressures from policies like REF. The authors recommend "education amongst wider geohazard community" as a solution. The authors need to explain what form of education they mean and provide evidence of how this will have the desired impact. Some of this arises later – maybe highlight that you will explore this more later in the article?

   In some cases, a change to policy may be needed rather than education. This could mean advocating for REF Impact Case Studies not being graded heavily on perceived global impact and rather explicitly valuing local impact that follows principles of good practice (e.g. DRR frameworks).

6. **"Training" as a means to ensure ethical data/information dissemination and clearly defining actions for particular actors:** As with Section 4.2, Section 4.4 and Section 5 advocates for "formal training around geohazard scientist involvement in ongoing disasters…". I agree in principle with this, but who should provide the training and take responsibility for ensuring that these principles are followed? Should educational institutions be embedding this in undergraduate education, post-graduate training and institutional-level training for faculty? Should it be professional bodies who run training, or grant-awarding bodies requiring ethical review for remote-sensing projects (I believe ERC already does this, for example). I recommend the authors attempt to more clearly map out the actions that different actors can take to meet their recommendations, if that is possible.

**Line-by-Line Review:**

Line 185: The mean cannot be larger than the maximum length of the interview (presumably should say max length was ~60 minutes).

Line 230: What is the significance of noting that transcription can waste time if the authors do not do anything about this? Did they retain the context-specific information that transcription lacks (e.g. body language)?

Line 310: A good example of the text being verbose in places "In further regard to the subject of geohazard remote sensing research being important in considering whether

to complete an ethical review…" could be shortened to "Ethical review is also important for …".

Line 346: "its" not "it's".

Line 516: "as soon as a University" is repeated.

Line 578: "desired impact strategy" is repeated.

Line 630: "…sight" not "…site".

Lines 630-635: Worth emphasising this point about feedback and tracking the impact of work being a challenge across many grant proposals. There are ways of working (e.g. theory of change models) that should be used to build feedback and impact estimation into projects.

Line 704: Consider re-phrasing start of the sentence – a little clunky.

Line 720-732: Seems to repeat some of the arguments from the section above regarding data volumes, processing power etc.

Line 901: Cite a source for an explanation of digital colonialism.

Line 913-919: Does this material belong in Section 4.2, as it seems to align with the theme of tension between research pressures and best practice?

Line 1037: spelling "… recognition is aligns…"

Line 1070: Would be better listing these topics as an actual bullet point list so they stand out more from the text. These seem to be key take-homes from the paper.

Line 1109: Double period at sentence end.